# Optimal Sketching for Kronecker Product Regression and Low Rank Approximation

Huaian Diao[*]   Rajesh Jayaram[†]   Zhao Song[‡]   Wen Sun[§]   David P. Woodruff[¶]

## Abstract

We study the Kronecker product regression problem, in which the design matrix is a Kronecker product of two or more matrices. Formally, given $A_i \in \mathbb{R}^{n_i \times d_i}$ for $i = 1, 2, \ldots, q$ where $n_i \gg d_i$ for each $i$, and $b \in \mathbb{R}^{n_1 n_2 \cdots n_q}$, let $\mathcal{A} = A_1 \otimes A_2 \otimes \cdots \otimes A_q$. Then for $p \in [1, 2]$, the goal is to find $x \in \mathbb{R}^{d_1 \cdots d_q}$ that approximately minimizes $\|\mathcal{A}x - b\|_p$. Recently, Diao, Song, Sun, and Woodruff (AISTATS, 2018) gave an algorithm which is faster than forming the Kronecker product $\mathcal{A} \in \mathbb{R}^{n_1 \cdots n_q \times d_1 \cdots d_q}$. Specifically, for $p = 2$ they achieve a running time of $O(\sum_{i=1}^{q} \mathrm{nnz}(A_i) + \mathrm{nnz}(b))$, where $\mathrm{nnz}(A_i)$ is the number of non-zero entries in $A_i$. Note that $\mathrm{nnz}(b)$ can be as large as $\Theta(n_1 \cdots n_q)$. For $p = 1$, $q = 2$ and $n_1 = n_2$, they achieve a worse bound of $O(n_1^{3/2} \mathrm{poly}(d_1 d_2) + \mathrm{nnz}(b))$.

In this work, we provide significantly faster algorithms. For $p = 2$, our running time is $O(\sum_{i=1}^{q} \mathrm{nnz}(A_i))$, which has no dependence on $\mathrm{nnz}(b)$. For $p < 2$, our running time is $O(\sum_{i=1}^{q} \mathrm{nnz}(A_i) + \mathrm{nnz}(b))$, which matches the prior best running time for $p = 2$. We also consider the related all-pairs regression problem, where given $A \in \mathbb{R}^{n \times d}, b \in \mathbb{R}^n$, we want to solve $\min_{x \in \mathbb{R}^d} \|\bar{A}x - \bar{b}\|_p$, where $\bar{A} \in \mathbb{R}^{n^2 \times d}, \bar{b} \in \mathbb{R}^{n^2}$ consist of all pairwise differences of the rows of $A, b$. We give an $O(\mathrm{nnz}(A))$ time algorithm for $p \in [1, 2]$, improving the $\Omega(n^2)$ time required to form $\bar{A}$. Finally, we initiate the study of Kronecker product low rank and low $t$-rank approximation. For input $\mathcal{A}$ as above, we give $O(\sum_{i=1}^{q} \mathrm{nnz}(A_i))$ time algorithms, which is much faster than computing $\mathcal{A}$.

## 1   Introduction

In the $q$-th order Kronecker product regression problem, one is given matrices $A_1, A_2, \ldots, A_q$, where $A_i \in \mathbb{R}^{n_i \times d_i}$, as well as a vector $b \in \mathbb{R}^{n_1 n_2 \cdots n_q}$, and the goal is to obtain a solution to the optimization problem:

$$\min_{x \in \mathbb{R}^{d_1 d_2 \cdots d_q}} \|(A_1 \otimes A_2 \cdots \otimes A_q)x - b\|_p,$$

---

[*]hadiao@nenu.edu.cn. Key Laboratory for Applied Statistics of MOE and School of Mathematics and Statistics, Northeast Normal University, China

[†]rkjayara@cs.cmu.edu. Carnegie Mellon University. Rajesh Jayaram would like to thank support from the Office of Naval Research (ONR) grant N00014-18-1-2562. This work was partly done while Rajesh Jayaram was visiting the Simons Institute for the Theory of Computing.

[‡]zhaosong@uw.edu. University of Washington. This work was partly done while Zhao Song was visiting the Simons Institute for the Theory of Computing.

[§]sun.wen@microsoft.com. Microsoft Research New York.

[¶]dwoodruf@cs.cmu.edu. Carnegie Mellon University. David Woodruff would like to thank support from the Office of Naval Research (ONR) grant N00014-18-1-2562. This work was also partly done while David Woodruff was visiting the Simons Institute for the Theory of Computing.

where $p \in [1, 2]$, and for a vector $x \in \mathbb{R}^n$ the $\ell_p$ norm is defined by $\|x\|_p = (\sum_{i=1}^n |x_i|^p)^{1/p}$. For $p = 2$, this is known as *least squares regression*, and for $p = 1$ this is known as *least absolute deviation regression*.

Kronecker product regression is a special case of ordinary regression in which the design matrix is highly structured. Namely, the design matrix is the Kronecker product of two or more smaller matrices. Such Kronecker product matrices naturally arise in applications such as spline regression, signal processing, and multivariate data fitting. We refer the reader to [VL92, VLP93, GVL13] for further background and applications of Kronecker product regression. As discussed in [DSSW18], Kronecker product regression also arises in structured blind deconvolution problems [OY05], and the bivariate problem of surface fitting and multidimensional density smoothing [EM06].

A recent work of Diao, Song, Sun, and Woodruff [DSSW18] utilizes *sketching* techniques to output an $x \in \mathbb{R}^{d_1 d_2 \cdots d_q}$ with objective function at most $(1 + \epsilon)$-times larger than optimal, for both least squares and least absolute deviation Kronecker product regression. Importantly, their time complexity is faster than the time needed to explicitly compute the product $A_1 \otimes \cdots \otimes A_q$. We note that sketching itself is a powerful tool for compressing extremely high dimensional data, and has been used in a number of tensor related problems, e.g., [SWZ16, LHW17, DSSW18, SWZ19b, AKK$^+$20].

For least squares regression, the algorithm of [DSSW18] achieves $O(\sum_{i=1}^q \mathrm{nnz}(A_i) + \mathrm{nnz}(b) + \mathrm{poly}(d/\epsilon))$ time, where $\mathrm{nnz}(C)$ for a matrix $C$ denotes the number of non-zero entries of $C$. Note that the focus is on the over-constrained regression setting, when $n_i \gg d_i$ for each $i$, and so the goal is to have a small running time dependence on the $n_i$'s. We remark that over-constrained regression has been the focus of a large body of work over the past decade, which primarily attempts to design fast regression algorithms in the big data (large sample size) regime, see, e.g., [Mah11, Woo14] for surveys.

Observe that explicitly forming the matrix $A_1 \otimes \cdots \otimes A_q$ would take $\prod_{i=1}^q \mathrm{nnz}(A_i)$ time, which can be as large as $\prod_{i=1}^q n_i d_i$, and so the results of [DSSW18] offer a large computational advantage. Unfortunately, since $b \in \mathbb{R}^{n_1 n_2 \cdots n_q}$, we can have $\mathrm{nnz}(b) = \prod_{i=1}^q n_i$, and therefore $\mathrm{nnz}(b)$ is likely to be the dominant term in the running time. This leaves open the question of whether it is possible to solve this problem in time *sub-linear* in $\mathrm{nnz}(b)$, with a dominant term of $O(\sum_{i=1}^q \mathrm{nnz}(A_i))$.

For least absolute deviation regression, the bounds of [DSSW18] achieved are still an improvement over computing $A_1 \otimes \cdots \otimes A_q$, though worse than the bounds for least squares regression. The authors focus on $q = 2$ and the special case $n = n_1 = n_2$. Here, they obtain a running time of $O(n^{3/2} \mathrm{poly}(d_1 d_2/\epsilon) + \mathrm{nnz}(b))$[6]. This leaves open the question of whether an *input-sparsity* $O(\mathrm{nnz}(A_1) + \mathrm{nnz}(A_2) + \mathrm{nnz}(b) + \mathrm{poly}(d_1 d_2/\epsilon))$ time algorithm exists.

**All-Pairs Regression**  In this work, we also study the related all-pairs regression problem. Given $A \in \mathbb{R}^{n \times d}, b \in \mathbb{R}^n$, the goal is to approximately solve the $\ell_p$ regression problem $\min_x \|\bar{A}x - \bar{b}\|_p$, where $\bar{A} \in \mathbb{R}^{n^2 \times d}$ is the matrix formed by taking all pairwise differences of the rows of $A$ (and $\bar{b}$ is defined similarly). For $p = 1$, this is known as the *rank regression estimator*, which has a long history in statistics. It is closely related to the renowned Wilconxon rank test [WL09], and enjoys the desirable property of being robust with substantial efficiency gain with respect to heavy-tailed random errors, while maintaining high efficiency for Gaussian errors [WKL09, WL09, WPB$^+$18, Wan19a]. In many ways, it has properties more desirable in practice than that of the Huber M-estimator [WPB$^+$18, Wan19b]. Recently, the all-pairs loss function was also used by [WPB$^+$18] as an alternative approach to overcoming the challenges of tuning parameter selection for the Lasso algorithm. However, the rank regression estimator is computationally intensive to compute, even for moderately sized data, since the standard procedure (for $p = 1$) is to solve a linear program with $O(n^2)$ constraints. In this work, we demonstrate the first highly efficient algorithm for this estimator.

**Low-Rank Approximation**  Finally, in addition to regression, we extend our techniques to the Low Rank Approximation (LRA) problem. Here, given a large data matrix $A$, the goal is to

find a low rank matrix $B$ which well-approximates $A$. LRA is useful in numerous applications, such as compressing massive datasets to their primary components for storage, denoising, and fast matrix-vector products. Thus, designing fast algorithms for approximate LRA has become a large and highly active area of research; see [Woo14] for a survey. For an incomplete list of recent work using sketching techniques for LRA, see [CW13, MM13, NN13, BW14, CW15b, CW15a, RSW16, BWZ16, SWZ17, MW17, CGK$^+$17, LHW17, SWZ18, BW18, SWZ19a, SWZ19b, SWZ19c, BBB$^+$19, IVWW19] and the references therein.

Motivated by the importance of LRA, we initiate the study of low-rank approximation of Kronecker product matrices. Given $q$ matrices $A_1, \cdots, A_q$ where $A_i \in \mathbb{R}^{n_i \times d_i}$, $n_i \gg d_i$, $A = \otimes_{i=1}^q A_i$, the goal is to output a rank-$k$ matrix $B \in \mathbb{R}^{n \times d}$ such that $\|B - A\|_F^2 \leq (1 + \epsilon) \operatorname{OPT}_k$, where $\operatorname{OPT}_k$ is the cost of the best rank-$k$ approximation, $n = n_1 \cdots n_q$, and $d = d_1 \cdots d_q$. Here $\|A\|_F^2 = \sum_{i,j} A_{i,j}^2$. The fastest general purpose algorithms for this problem run in time $O(\operatorname{nnz}(A) + \operatorname{poly}(dk/\epsilon))$ [CW13]. However, as in regression, if $A = \otimes_{i=1}^q A_i$, we have $\operatorname{nnz}(A) = \prod_{i=1}^q \operatorname{nnz}(A_i)$, which grows very quickly. Instead, one might also hope to obtain a running time of $O(\sum_{i=1}^q \operatorname{nnz}(A_i) + \operatorname{poly}(dk/\epsilon))$.

## 1.1   Our Contributions

Our main contribution is an input sparsity time $(1 + \epsilon)$-approximation algorithm to Kronecker product regression for every $p \in [1, 2]$, and $q \geq 2$. Given $A_i \in \mathbb{R}^{n_i \times d_i}$, $i = 1, \ldots, q$, and $b \in \mathbb{R}^n$ where $n = \prod_{i=1}^q n_i$, together with accuracy parameter $\epsilon \in (0, 1/2)$ and failure probability $\delta > 0$, the goal is to output a vector $x' \in \mathbb{R}^d$ where $d = \prod_{i=1}^q d_i$ such that $\|(A_1 \otimes \cdots \otimes A_q)x' - b\|_p \leq (1 + \epsilon) \min_x \|(A_1 \otimes \cdots \otimes A_q)x - b\|_p$ holds with probability at least $1 - \delta$. For $p = 2$, our algorithm runs in $\widetilde{O}\left(\sum_{i=1}^q \operatorname{nnz}(A_i)\right) + \operatorname{poly}(d\delta^{-1}/\epsilon))$ time.[7] Notice that this is *sub-linear* in the input size, since it does not depend on $\operatorname{nnz}(b)$. For $p < 2$, the running time is $\widetilde{O}\left((\sum_{i=1}^q \operatorname{nnz}(A_i) + \operatorname{nnz}(b) + \operatorname{poly}(d/\epsilon)) \log(1/\delta)\right)$.

Observe that in both cases, this running time is significantly faster than the time to write down $A_1 \otimes \cdots \otimes A_q$. For $p = 2$, up to logarithmic factors, the running time is the same as the time required to simply read each of the $A_i$. Moreover, in the setting $p < 2$, $q = 2$ and $n_1 = n_2$ considered in [DSSW18], our algorithm offers a substantial improvement over their running time of $O(n^{3/2} \operatorname{poly}(d_1 d_2/\epsilon))$. We empirically evaluate our Kronecker product regression algorithm on exactly the same datasets as those used in [DSSW18]. For $p \in \{1, 2\}$, the accuracy of our algorithm is nearly the same as that of [DSSW18], while the running time is significantly faster.

For the all-pairs (or rank) regression problem, we first note that for $A \in \mathbb{R}^{n \times d}$, one can rewrite $\bar{A} \in \mathbb{R}^{n^2 \times d}$ as the difference of Kronecker products $\bar{A} = A \otimes \mathbf{1}^n - \mathbf{1}^n \otimes A$ where $\mathbf{1}^n \in \mathbb{R}^n$ is the all ones vector. Since $\bar{A}$ is not a Kronecker product itself, our earlier techniques for Kronecker product regression are not directly applicable. Therefore, we utilize new ideas, in addition to careful sketching techniques, to obtain an $\widetilde{O}(\operatorname{nnz}(A) + \operatorname{poly}(d/\epsilon))$ time algorithm for $p \in [1, 2]$, which improves substantially on the $O(n^2 d)$ time required to even compute $\bar{A}$, by a factor of at least $n$.

Our main technical contribution for both our $\ell_p$ regression algorithm and the rank regression problem is a novel and highly efficient $\ell_p$ sampling algorithm. Specifically, for the rank-regression problem we demonstrate, for a given $x \in \mathbb{R}^d$, how to independently sample $s$ entries of a vector $\bar{A}x = y \in \mathbb{R}^{n^2}$ from the $\ell_p$ distribution $(|y_1|^p/\|y\|_p^p, \ldots, |y_{n^2}|^p/\|y\|_p^p)$ in $\widetilde{O}(nd + \operatorname{poly}(ds))$ time. For the $\ell_p$ regression problem, we demonstrate the same result when $y = (A_1 \otimes \cdots \otimes A_q)x - b \in \mathbb{R}^{n_1 \cdots n_q}$, and in time $\widetilde{O}(\sum_{i=1}^q \operatorname{nnz}(A_i) + \operatorname{nnz}(b) + \operatorname{poly}(ds))$. This result allows us to sample a small number of rows of the input to use in our sketch. Our algorithm draws from a large number of disparate sketching techniques, such as the dyadic trick for quickly finding heavy hitters [CM05, KNPW11, LNNT16, NS19], and the precision sampling framework from the streaming literature [AKO11].

For the Kronecker Product Low-Rank Approximation (LRA) problem, we give an input sparsity $O(\sum_{i=1}^q \operatorname{nnz}(A_i) + \operatorname{poly}(dk/\epsilon))$-time algorithm which computes a rank-$k$ matrix $B$ such that $\|B - \otimes_{i=1}^q A_i\|_F^2 \leq (1 + \epsilon) \min_{\operatorname{rank} - k\ B'} \|B' - \otimes_{i=1}^q A_i\|_F^2$. Note again that the dominant term $\sum_{i=1}^q \operatorname{nnz}(A_i)$ is substantially smaller than the $\operatorname{nnz}(A) = \prod_{i=1}^q \operatorname{nnz}(A_i)$ time required to write

down the Kronecker Product $A$, which is also the running time of state-of-the-art general purpose LRA algorithms [CW13, MM13, NN13]. Thus, our results demonstrate that substantially faster algorithms for approximate LRA are possible for inputs with a Kronecker product structure.

Finally, motivated by [VL00], we use our techniques to solve the low-trank approximation problem, where we are given an arbitrary matrix $A \in \mathbb{R}^{n^q \times n^q}$, and the goal is to output a trank-$k$ matrix $B \in \mathbb{R}^{n^q \times n^q}$ such that $\|B - A\|_F$ is minimized. Here, the trank of a matrix $B$ is the smallest integer $k$ such that $B$ can be written as a summation of $k$ matrices, where each matrix is the Kronecker product of $q$ matrices with dimensions $n \times n$. Compressing a matrix $A$ to a low-trank approximation yields many of the same benefits as LRA, such as compact representation, fast matrix-vector product, and fast matrix multiplication, and thus is applicable in many of the settings where LRA is used. Using similar sketching ideas, we provide an $O(\sum_{i=1}^q \mathrm{nnz}(A_i) + \mathrm{poly}(d_1 \cdots d_q/\epsilon))$ time algorithm for this problem under various loss functions. Our results for low-trank approximation can be found in the full version of this work.

## 2 Preliminaries

**Notation**  For a tensor $A \in \mathbb{R}^{n_1 \times n_2 \times n_3}$, we use $\|A\|_p$ to denote the entry-wise $\ell_p$ norm of $A$, i.e., $\|A\|_p = (\sum_{i_1} \sum_{i_2} \sum_{i_3} |A_{i_1,i_2,i_3}|^p)^{1/p}$. For $n \in \mathbb{N}$, let $[n] = \{1, 2, \ldots, n\}$. For a matrix $A$, let $A_{i,*}$ denote the $i$-th row of $A$, and $A_{*,j}$ the $j$-th column. For $a, b \in \mathbb{R}$ and $\epsilon \in (0, 1)$, we write $a = (1 \pm \epsilon)b$ to denote $(1 - \epsilon)b \leq a \leq (1 + \epsilon)b$. We now define various sketching matrices used by our algorithms.

**Stable Transformations**  We will utilize the well-known $p$-stable distribution, $\mathcal{D}_p$ (see [Nol07, Ind06] for further discussion), which exist for $p \in (0, 2]$. For $p \in (0, 2)$, $X \sim \mathcal{D}_p$ is defined by its characteristic function $\mathbb{E}_X[\exp(\sqrt{-1}tX)] = \exp(-|t|^p)$, and can be efficiently generated to a fixed precision [Nol07, KNW10]. For $p = 2$, $\mathcal{D}_2$ is just the standard Gaussian distribution, and for $p = 1$, $\mathcal{D}_1$ is the *Cauchy* distribution. The distribution $\mathcal{D}_p$ has the property that if $z_1, \ldots, z_n \sim D_p$ are i.i.d., and $a \in \mathbb{R}^n$, then $\sum_{i=1}^n z_i a_i \sim z\|a\|_p$ where $\|a\|_p = (\sum_{i=1}^n |a_i|^p)^{1/p}$, and $z \sim \mathcal{D}_p$. This property will allow us to utilize sketches with entries independently drawn from $\mathcal{D}_p$ to preserve the $\ell_p$ norm.

**Definition 2.1** (Dense $p$-stable Transform, [CDMI$^+$13, SW11]). *Let $p \in [1, 2]$. Let $S = \sigma \cdot C \in \mathbb{R}^{m \times n}$, where $\sigma$ is a scalar, and each entry of $C \in \mathbb{R}^{m \times n}$ is chosen independently from $\mathcal{D}_p$.*

We will also need a sparse version of the above.

**Definition 2.2** (Sparse $p$-Stable Transform, [MM13, CDMI$^+$13]). *Let $p \in [1, 2]$. Let $\Pi = \sigma \cdot SC \in \mathbb{R}^{m \times n}$, where $\sigma$ is a scalar, $S \in \mathbb{R}^{m \times n}$ has each column chosen independently and uniformly from the $m$ standard basis vectors of $\mathbb{R}^m$, and $C \in \mathbb{R}^{n \times n}$ is a diagonal matrix with diagonals chosen independently from the standard $p$-stable distribution. For any matrix $A \in \mathbb{R}^{n \times d}$, $\Pi A$ can be computed in $O(\mathrm{nnz}(A))$ time.*

One nice property of $p$-stable transformations is that they provide *low-distortion $\ell_p$ embeddings*.

**Lemma 2.3** (Theorem 1.4 of [WW19]; see also Theorem 2 and 4 of [MM13] for earlier work [8] ). *Fix $A \in \mathbb{R}^{n \times d}$, and let $S \in \mathbb{R}^{k \times n}$ be a sparse or dense $p$-stable transform for $p \in [1, 2)$, with $k = \Theta(d^2/\delta)$. Then with probability $1 - \delta$, for all $x \in \mathbb{R}^d$:*

$$\|Ax\|_p \leq \|SAx\|_p \leq O(d \log d)\|Ax\|_p$$

We simply call a matrix $S \in \mathbb{R}^{k \times n}$ a low distortion $\ell_p$ embedding for $A \in \mathbb{R}^{n \times d}$ if it satisfies the above inequality for all $x \in \mathbb{R}^d$.

**Leverage Scores & Well Condition Bases.**  We now introduce the notions of $\ell_2$ leverage scores and well-conditioned bases for a matrix $A \in \mathbb{R}^{n \times d}$.

**Definition 2.4** ($\ell_2$-Leverage Scores, [Woo14, BSS12]). *Given a matrix $A \in \mathbb{R}^{n \times d}$, let $A = Q \cdot R$ denote the QR factorization of matrix $A$. For each $i \in [n]$, we define $\sigma_i = \frac{\|(AR^{-1})_i\|_2^2}{\|AR^{-1}\|_F^2}$, where*

$(AR^{-1})_i \in \mathbb{R}^d$ is the $i$-th row of matrix $(AR^{-1}) \in \mathbb{R}^{n \times d}$. We say that $\sigma \in \mathbb{R}^n$ is the $\ell_2$ leverage score vector of $A$.

**Definition 2.5** $((\ell_p, \alpha, \beta)$ Well-Conditioned Basis, [Cla05])**.** *Given a matrix $A \in \mathbb{R}^{n \times d}$, we say $U \in \mathbb{R}^{n \times d}$ is an $(\ell_p, \alpha, \beta)$ well-conditioned basis for the column span of $A$ if the columns of $U$ span the columns of $A$, and if for any $x \in \mathbb{R}^d$, we have $\alpha \|x\|_p \leq \|Ux\|_p \leq \beta \|x\|_p$, where $\alpha \leq 1 \leq \beta$. If $\beta/\alpha = d^{O(1)}$, then we simply say that $U$ is an $\ell_p$ well conditioned basis for $A$.*

**Fact 2.6** ([WW19, MM13])**.** *Let $A \in \mathbb{R}^{n \times d}$, and let $SA \in \mathbb{R}^{k \times d}$ be a low distortion $\ell_p$ embedding for $A$ (see Lemma 2.3), where $k = O(d^2/\delta)$. Let $SA = QR$ be the QR decomposition of $SA$. Then $AR^{-1}$ is an $\ell_p$ well-conditioned basis with probability $1 - \delta$.*

---

**Algorithm 1** Our $\ell_2$ Kronecker Product Regression Algorithm

---

1: **procedure** $\ell_2$ KRONECKER REGRESSION$(( \{A_i, n_i, d_i\}_{i \in [q]}, b))$ &emsp;&emsp;&emsp;&emsp; ▷ Theorem 3.1
2: &emsp; $d \leftarrow \prod_{i=1}^q d_i$, $n \leftarrow \prod_{i=1}^q n_i$, $m \leftarrow \Theta(d/(\delta\epsilon^2))$.
3: &emsp; Compute approximate leverage scores $\widetilde{\sigma}_i(A_j)$ for all $j \in [q]$, $i \in [n_j]$.
4: &emsp; Construct diagonal leverage score sampling matrix $D \in \mathbb{R}^{n \times n}$, with $m$ non-zero entries
5: &emsp; Compute (via the psuedo-inverse)
6: &emsp;&emsp;&emsp;&emsp;&emsp;&emsp; $\widehat{x} = \arg\min_{x \in \mathbb{R}^d} \|D(A_1 \otimes A_2 \otimes \cdots \otimes A_q)x - Db\|_2$
7: &emsp; **return** $\widehat{x}$
8: **end procedure**

---

**Algorithm 2** Our $\ell_p$ Kronecker Product Regression Algorithm, $1 \leq p < 2$

---

1: **procedure** $O(1)$-APPROXIMATE $\ell_p$ REGRESSION$(\{A_i, n_i, d_i\}_{i \in [q]})$ &emsp;&emsp; ▷ Theorem 3.2
2: &emsp; $d \leftarrow \prod_{i=1}^q d_i$, $n \leftarrow \prod_{i=1}^q n_i$.
3: &emsp; **for** $i = 1, \ldots, q$ **do**
4: &emsp;&emsp; $s_i \leftarrow O(qd_i^2)$
5: &emsp;&emsp; Generate sparse $p$-stable transform $S_i \in \mathbb{R}^{s_i \times n}$ (def 2.2) &emsp;&emsp;&emsp; ▷ Lemma 2.3
6: &emsp;&emsp; Take the QR factorization of $S_i A_i = Q_i R_i$ to obtain $R_i \in \mathbb{R}^{d_i \times d_i}$ &emsp;&emsp; ▷ Fact 2.6
7: &emsp;&emsp; Let $Z \in \mathbb{R}^{d \times \tau}$ be a dense $p$-stable transform for $\tau = \Theta(\log(n))$ &emsp; ▷ Definition 2.1
8: &emsp;&emsp; **for** $j = 1, \ldots, n_i$ **do**
9: &emsp;&emsp;&emsp; $a_{i,j} \leftarrow \text{median}_{\eta \in [\tau]} \{(|(A_i R_i^{-1} Z)_{j,\eta}|/\theta_p)^p\}$, where $\theta_p$ is the median of $\mathcal{D}_p$.
10: &emsp;&emsp; **end for**
11: &emsp; **end for**
12: &emsp; Define a distribution $\mathcal{D} = \{q_1', q_1', \ldots, q_n'\}$ by $q'_{\sum_{i=1}^q j_i \prod_{l=1}^{j-1} n_l} = \prod_{i=1}^q a_{i,j_i}$.
13: &emsp; Let $\Pi \in \mathbb{R}^{n \times n}$ denote a diagonal sampling matrix, where $\Pi_{i,i} = 1/q_i^{1/p}$ with probability $q_i = \min\{1, r_1 q_i'\}$ and 0 otherwise, where $r_1 = \Theta(d^3/\epsilon^2)$. &emsp;&emsp;&emsp;&emsp; ▷ [DDH+09]
14: &emsp; Let $x' \in \mathbb{R}^d$ denote the solution of
15: &emsp;&emsp;&emsp;&emsp; $\min_{x \in \mathbb{R}^d} \|\Pi(A_1 \otimes A_2 \otimes \cdots \otimes A_q)x - \Pi b\|_p$
16: &emsp; **return** $x'$
17: **end procedure**
18: **procedure** $(1 + \epsilon)$-APPROXIMATE $\ell_p$ REGRESSION$(x' \in \mathbb{R}^d)$
19: &emsp; Implicitly define $\rho = (A_1 \otimes A_2 \otimes \cdots \otimes A_q)x' - b \in \mathbb{R}^n$
20: &emsp; Compute a diagonal sampling matrix $\Sigma \in \mathbb{R}^{n \times n}$ such that $\Sigma_{i,i} = 1/\alpha_i^{1/p}$ with probability $\alpha_i = \min\{1, \max\{q_i, r_2 |\rho_i|^p/\|\rho\|_p^p\}\}$ where $r_2 = \Theta(d^3/\epsilon^3)$.
21: &emsp; Compute $\widehat{x} = \arg\min_{x \in \mathbb{R}^d} \|\Sigma(A_1 \otimes A_2 \otimes \cdots \otimes A_q) - \Sigma b\|_p$ (via convex optimization methods, e.g., [BCLL18, AKPS19, LSZ19])
22: &emsp; **return** $\widehat{x}$
23: **end procedure**

---

# 3 Kronecker Product Regression

We first introduce our algorithm for $p = 2$. Our algorithm for $1 \leq p < 2$ is given in Section 3.1. Our regression algorithm for $p = 2$ is formally stated in Algorithm 1. Recall that our input design matrix is $A = \otimes_{i=1}^q A_i$, where $A_i \in \mathbb{R}^{n_i \times d_i}$, and we are also given $b \in \mathbb{R}^{n_1 \cdots n_q}$. Let

$n = \prod_{i=1}^{q} n_i$ and $d = \prod_{i=1}^{q} d_i$. The crucial insight of the algorithm is that one can approximately compute the leverage scores of $A$ given only good approximations to the leverage scores of each $A_i$. Applying this fact gives a efficient algorithm for sampling rows of $A$ with probability proportional to the leverage scores. Following standard arguments, we will show that by restricting the regression problem to the sampled rows, we can obtain our desired $(1 \pm \epsilon)$-approximate solution efficiently. Our main theorem for this section is stated below.

**Theorem 3.1** (Kronecker product $\ell_2$ regression)**.** *Let $D \in \mathbb{R}^{n \times n}$ be the diagonal row sampling matrix generated in Algorithm 1, with $m = \Theta(d/(\delta\epsilon^2))$ non-zero entries, and let $A = \otimes_{i=1}^{q} A_i$, where $A_i \in \mathbb{R}^{n_i \times d_i}$, and $b \in \mathbb{R}^n$, where $n = \prod_{i=1}^{q} n_i$ and $d = \prod_{i=1}^{q} d_i$. Then let $\widehat{x} = \arg\min_{x \in \mathbb{R}^d} \|DAx - Db\|_2$, and let $x^* = \arg\min_{x' \in \mathbb{R}^d} \|Ax - b\|_2$. Then with probability $1 - \delta$, we have $\|A\widehat{x} - b\|_2 \le (1 + \epsilon)\|Ax^* - b\|_2$. Moreover, the total running time required to compute $\widehat{x}$ is $\widetilde{O}(\sum_{i=1}^{q} \mathrm{nnz}(A_i) + (dq/(\delta\epsilon))^{O(1)})$. [9]*

### 3.1 Kronecker Product $\ell_p$ Regression

We now consider $\ell_p$ regression for $1 \le p < 2$. Our algorithm is stated formally in Algorithm 2. Our main theorem is as follows.

**Theorem 3.2** (Main result, $\ell_p$ $(1+\epsilon)$-approximate regression)**.** *Fix $1 \le p < 2$. Then for any constant $q = O(1)$, given matrices $A_1, A_2, \cdots, A_q$, where $A_i \in \mathbb{R}^{n_i \times d_i}$, let $n = \prod_{i=1}^{q} n_i$, $d = \prod_{i=1}^{q} d_i$. Let $\widehat{x} \in \mathbb{R}^d$ be the output of Algorithm 2. Then*

$$\|(A_1 \otimes A_2 \otimes \cdots \otimes A_q)\widehat{x} - b\|_p \le (1 + \epsilon) \min_{x \in \mathbb{R}^n} \|(A_1 \otimes A_2 \otimes \cdots \otimes A_q)x - b\|_p$$

*holds with probability at least $1 - \delta$. In addition, our algorithm takes $\widetilde{O}\left((\sum_{i=1}^{q} \mathrm{nnz}(A_i) + \mathrm{nnz}(b) + \mathrm{poly}(d \log(1/\delta)/\epsilon)) \log(1/\delta)\right)$ time to output $\widehat{x} \in \mathbb{R}^d$.*

Our high level approach follows that of [DDH+09]. Namely, we first obtain a vector $x'$ which is an $O(1)$-approximate solution to the optimal solution. This is done by first constructing (implicitly) a matrix $U \in \mathbb{R}^{n \times d}$ that is a well-conditioned basis for the design matrix $A_1 \otimes \cdots \otimes A_q$. We then efficiently sample rows of $U$ with probability proportional to their $\ell_p$ norm (which must be done without even explicitly computing most of $U$). We then use the results of [DDH+09] to demonstrate that solving the regression problem constrained to these sampled rows gives a solution $x' \in \mathbb{R}^d$ such that $\|(A_1 \otimes \cdots \otimes A_q)x' - b\|_p \le 8 \min_{x \in \mathbb{R}^d} \|(A_1 \otimes \cdots \otimes A_q)x' - b\|_p$.

We define the *residual error* $\rho = (A_1 \otimes \cdots \otimes A_q)x' - b \in \mathbb{R}^n$ of $x'$. Our goal is to sample additional rows $i \in [n]$ with probability proportional to their residual error $|\rho_i|^p/\|\rho\|_p^p$, and solve the regression problem restricted to the sampled rows. However, we cannot afford to compute even a small fraction of the entries in $\rho$ (even when $b$ is dense, and certainly not when $b$ is sparse). So to carry out this sampling efficiently, we design an involved, multi-part sketching and sampling routine. This sampling technique is the main technical contribution of this section, and relies on a number of techniques, such as the Dyadic trick for quickly finding heavy hitters from the streaming literature, and a careful pre-processing step to avoid a $\mathrm{poly}(d)$-blow up in the runtime. Given these samples, we can obtain the solution $\widehat{x}$ after solving the regression problem on the sampled rows, and the fact that this gives a $(1 + \epsilon)$ approximate solution will follow from Theorem 6 of [DDH+09].

## 4 All-Pairs Regression

Given a matrix $A \in \mathbb{R}^{n \times d}$ and $b \in \mathbb{R}^n$, let $\bar{A} \in \mathbb{R}^{n^2 \times d}$ be the matrix such that $\bar{A}_{i+(j-1)n,*} = A_{i,*} - A_{j,*}$, and let $\bar{b} \in \mathbb{R}^{n^2}$ be defined by $\bar{b}_{i+(j-1)n} = b_i - b_j$. Thus, $\bar{A}$ consists of all pairwise differences of rows of $A$, and $\bar{b}$ consists of all pairwise differences of rows of $b$,. The $\ell_p$ all pairs regression problem on the inputs $A, b$ is to solve $\min_{x \in \mathbb{R}^d} \|\bar{A}x - \bar{b}\|_p$.

First note that this problem has a close connection to Kronecker product regression. Namely, the matrix $\bar{A}$ can be written $\bar{A} = A \otimes \mathbf{1}^n - \mathbf{1}^n \otimes A$, where $\mathbf{1}^n \in \mathbb{R}^n$ is the all 1's vector. Similarly, $\bar{b} = b \otimes \mathbf{1}^n - \mathbf{1}^n \otimes b$. For simplicity, we now drop the superscript and write $\mathbf{1} = \mathbf{1}^n$.

Our algorithm is given formally in Algorithm 3. The main technical step takes place on line 7, where we sample rows of the matrix $(F \otimes \mathbf{1} - \mathbf{1} \otimes F)R^{-1}$ with probability proportional to their $\ell_p$ norms. This is done by an involved sampling procedure described in the full version of this work. We summarize the guarantee of our algorithm in the following theorem.

**Theorem 4.1.** *Given $A \in \mathbb{R}^{n \times d}$ and $b \in \mathbb{R}^n$, for $p \in [1, 2]$, let $\bar{A} = A \otimes \mathbf{1} - \mathbf{1} \otimes A \in \mathbb{R}^{n^2 \times d}$ and $\bar{b} = b \otimes \mathbf{1} - \mathbf{1} \otimes b \in \mathbb{R}^{n^2}$. Then there is an algorithm for that outputs $\widehat{x} \in \mathbb{R}^d$ such that with probability $1 - \delta$ we have $\|\bar{A}\widehat{x} - \bar{b}\|_p \le (1 + \epsilon) \min_{x \in \mathbb{R}^d} \|\bar{A}x - \bar{b}\|_p$. The running time is $\widetilde{O}(\mathrm{nnz}(A) + (d/(\epsilon\delta))^{O(1)})$.*

---

**Algorithm 3** Our All-Pairs Regression Algorithm

1: **procedure** ALL-PAIRS REGRESSION($A, b$)
2:     $F = [A, b] \in \mathbb{R}^{n \times d+1}$. $r \leftarrow \mathrm{poly}(d/\epsilon)$
3:     Generate $S_1, S_2 \in \mathbb{R}^{k \times n}$ sparse $p$-stable transforms for $k = \mathrm{poly}(d/(\epsilon\delta))$.
4:     Sketch $(S_1 \otimes S_2)(F \otimes \mathbf{1} - \mathbf{1} \otimes F)$.
5:     Compute $QR$ decomposition: $(S_1 \otimes S_2)(F \otimes \mathbf{1} - \mathbf{1} \otimes F) = QR$.
6:     Let $M = (F \otimes \mathbf{1} - \mathbf{1} \otimes F)R^{-1}$, and $\sigma_i = \|M_{i,*}\|_p^p / \|M\|_p^p$.
7:     Obtain row sampling diagonal matrix $\Pi \in \mathbb{R}^{n \times n}$ such that $\Pi_{i,i} = 1/\widetilde{q}_i^{1/p}$ independently with probability $q_i \ge \min\{1, r\sigma_i\}$, where $\widetilde{q}_i = (1 \pm \epsilon^2)q_i$.
8:     **return** $\widehat{x}$, where $\widehat{x} = \arg\min_{x \in \mathbb{R}^d} \|\Pi(\bar{A}x - \bar{b})\|_p$.
9: **end procedure**

---

## 5 Low Rank Approximation of Kronecker Product Matrices

We now consider low rank approximation of Kronecker product matrices. Given $q$ matrices $A_1, A_2, \ldots, A_q$, where $A_i \in \mathbb{R}^{n_i \times d_i}$, the goal is to output a rank-$k$ matrix $B \in \mathbb{R}^{n \times d}$, where $n = \prod_{i=1}^q n_i$ and $d = \prod_{i=1}^q d_i$, such that $\|B - A\|_F \le (1 + \epsilon) \mathrm{OPT}_k$, where $\mathrm{OPT}_k = \min_{\mathrm{rank}-k \, A'} \|A' - A\|_F$, and $A = \otimes_{i=1}^q A_i$. Our approach employs the Count-Sketch distribution of matrices [CW13, Woo14]. A count-sketch matrix $S$ is generated as follows. Each column of $S$ contains exactly one non-zero entry. The non-zero entry is placed in a uniformly random row, and the value of the non-zero entry is either $1$ or $-1$ chosen uniformly at random.

Our algorithm is as follows. We sample $q$ independent Count-Sketch matrices $S_1, \ldots S_q$, with $S_i \in \mathbb{R}^{k_i \times n_i}$, where $k_1 = \cdots = k_q = \Theta(qk^2/\epsilon^2)$. We then compute $M = (\otimes_{i=1}^q S_i)A$, and let $U \in \mathbb{R}^{k \times d}$ be the top $k$ right singular vectors of $M$. Finally, we output $B = AU^\top U$ in factored form (as $q + 1$ separate matrices, $A_1, A_2, \ldots, A_q, U$), as the desired rank-$k$ approximation to $A$. The following theorem demosntrates the correctness of this algorithm.

**Theorem 5.1.** *For any constant $q \ge 2$, there is an algorithm which runs in time $O((\sum_{i=1}^q \mathrm{nnz}(A_i) + d\,\mathrm{poly}(k/\epsilon))\log(1/\delta))$ and outputs a rank $k$-matrix $B$ in factored form such that $\|B - A\|_F \le (1 + \epsilon)\mathrm{OPT}_k$ with probability $1 - \delta$. with probability $9/10$.*

## 6 Numerical Simulations

In our numerical simulations, we compare our algorithms to two baselines: (1) brute force, i.e., directly solving regression without sketching, and (2) the methods based sketching developed in [DSSW18]. All methods were implemented in Matlab on a Linux machine. We remark that in our implementation, we simplified some of the steps of our theoretical algorithm, such as the residual sampling algorithm used in Alg. 2. We found that in practice, even with these simplifications, our algorithms already demonstrated substantial improvements over prior work.

Following the experimental setup in [DSSW18], we generate matrices $A_1 \in \mathbb{R}^{300 \times 15}$, $A_2 \in \mathbb{R}^{300 \times 15}$, and $b \in \mathbb{R}^{300^2}$, such that all entries of $A_1, A_2, b$ are sampled i.i.d. from a normal distribution. Note that $A_1 \otimes A_2 \in \mathbb{R}^{90000 \times 225}$. We define $T_{\mathrm{bf}}$ to be the time of the brute force algorithm,

Table 1: Results for $\ell_2$ and $\ell_1$-regression with respect to different sketch sizes $m$.

|  | $m$ | $m/n$ | $r_e$ | $r'_e$ | $r_t$ | $r'_t$ |
|---|---|---|---|---|---|---|
| $\ell_2$ | 8100 | .09 | 2.48% | 1.51% | 0.05 | 0.22 |
|  | 12100 | .13 | 1.55% | 0.98% | 0.06 | 0.24 |
|  | 16129 | .18 | 1.20% | 0.71% | 0.07 | 0.08 |
| $\ell_1$ | 2000 | .02 | 7.72% | 9.10% | 0.02 | 0.59 |
|  | 4000 | .04 | 4.26% | 4.00% | 0.03 | 0.75 |
|  | 8000 | .09 | 1.85% | 1.6% | 0.07 | 0.83 |
|  | 12000 | .13 | 1.29% | 0.99% | 0.09 | 0.79 |
|  | 16000 | .18 | 1.01% | 0.70% | 0.14 | 0.90 |

$T_{\text{old}}$ to be the time of the algorithms from [DSSW18], and $T_{\text{ours}}$ to be the time of our algorithms. We are interested in the time ratio with respect to the brute force algorithm and the algorithms from [DSSW18], defined as, $r_t = T_{\text{ours}}/T_{\text{bf}}$, and $r'_t = T_{\text{ours}}/T_{\text{old}}$. The goal is to show that our methods are significantly faster than both baselines, i.e., both $r_t$ and $r'_t$ are significantly less than $1$.

We are also interested in the quality of the solutions computed from our algorithms, compared to the brute force method and the method from [DSSW18]. Denote the solution from our method as $x_{\text{our}}$, the solution from the brute force method as $x_{\text{bf}}$, and the solution from the method in [DSSW18] as $x_{\text{old}}$. We define the relative residual percentage $r_e$ and $r'_e$ to be:

$$r_e = 100 \frac{|\|\mathcal{A}x_{\text{ours}} - b\| - \|\mathcal{A}x_{\text{bf}} - b\||}{\|\mathcal{A}x_{\text{bf}} - b\|}, \quad r'_e = 100 \frac{|\|\mathcal{A}x_{\text{old}} - b\| - \|\mathcal{A}x_{\text{bf}} - b\||}{\|\mathcal{A}x_{\text{bf}} - b\|}$$

Where $\mathcal{A} = A_1 \otimes A_2$. The goal is to show that $r_e$ is close zero, i.e., our approximate solution is comparable to the optimal solution in terms of minimizing the error $\|\mathcal{A}x - b\|$.

Throughout the simulations, we use a moderate input matrix size so that we can accommodate the brute force algorithm and to compare to the exact solution. We consider varying values of $m$, where $M$ denotes the size of the sketch (number of rows) used in either the algorithms of [DSSW18] or the algorithms in this paper. We also include a column $m/n$ in the table, which is the ratio between the size of the sketch and the original matrix $A_1 \otimes A_2$. Note in this case that $n = 90000$.

**Simulation Results for $\ell_2$** We first compare our algorithm, Alg. 1, to baselines under the $\ell_2$ norm. In our implementation, $\min_x \|Ax - b\|_2$ is solved by Matlab backslash $A \backslash b$. Table 1 summarizes the comparison between our approach and the two baselines. The numbers are averaged over 5 random trials. First of all, we notice that our method in general provides slightly less accurate solutions than the method in [DSSW18], i.e., $r_e > r'_e$ in this case. However, comparing to the brute force algorithm, our method still generates relatively accurate solutions, especially when $m$ is large, e.g., the relative residual percentage w.r.t. the optimal solution is around $1\%$ when $m \approx 16000$. On the other hand, as suggested by our theoretical improvements for $\ell_2$, our method is significantly faster than the method from [DSSW18], consistently across all sketch sizes $m$. Note that when $m \approx 16000$, our method is around 10 times faster than the method in [DSSW18]. For small $m$, our approach is around 5 times faster than the method in [DSSW18].

**Simulation Results for $\ell_1$** We compare our algorithm, Alg. 2, to two baselines under the $\ell_1$-norm. The first is a brute-force solution, and the second is the algorithm for [DSSW18]. For $\min_x \|Ax - b\|_1$, the brute for solution is obtained via a Linear Programming solver in Gurobi [GO16]. Table 1 summarizes the comparison of our approach to the two baselines under the $\ell_1$-norm. The statistics are averaged over 5 random trials. Compared to the Brute Force algorithm, our method is consistently around 10 times faster, while in general we have relative residual percentage around $1\%$. Compared to the method from [DSSW18], our approach is consistently faster (around 1.3 times faster). Note our method has slightly higher accuracy than the one from [DSSW18] when the sketch size is small, but slightly worse accuracy when the sketch size increases.

## Acknowledgments

The authors would like to thank Lan Wang and Ruosong Wang for a helpful discussion. The authors would like to thank Lan Wang for introducing the All-Pairs Regression problem to us.

## Footnotes

[6]We remark that while the $\mathrm{nnz}(b)$ term is not written in the Theorem of [DSSW18], their approach of leverage score sampling from a well-conditioned basis requires one to sample from a well conditioned basis of $[A_1 \otimes A_2, b]$ for a subspace embedding. As stated, their algorithm only sampled from $[A_1 \otimes A_2]$. To fix this omission, their algorithm would require an additional $\mathrm{nnz}(b)$ time to leverage score sample from the augmented matrix.

[7]For a function $f(n, d, \epsilon, \delta)$, $\widetilde{O}(f) = O(f \cdot \operatorname{poly}(\log n))$

[8]In discussion with the authors of these works, the original $O((d \log d)^{1/p})$ distortion factors stated in these papers should be replaced with $O(d \log d)$; as we do not optimize the $\mathrm{poly}(d)$ factors in our analysis, this does not affect our bounds.

[9] We remark that the exponent of $d$ in the runtime can be bounded by 3. To see this, first note that the main computation taking place is the leverage score computation. For a $q$ input matrices, we need to generate the leverage scores to precision $\Theta(1/q)$, and the complexity to achieve this is $O(d^3/q^4)$ by the results of [CW13]. The remaining computation is to compute the pseudo-inverse of a $d/\epsilon^2 \times d$ matrix, which requires $O(d^3/\epsilon^2)$ time, so the additive term in the Theorem can be replaced with $O(d^3/\epsilon^2 + d^3/q^4)$.

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
