[Supplementary Material · kronecker_supp.pdf]

## Appendix

## A   Missing Proofs from Section 3

In this section, we prove correctness of our $\ell_2$ Kronecker product regression algorithm. Specifically, we prove Theorem 3.1. To prove correctness, we need to establish several facts about the leverage scores of a Kronecker product.

**Proposition A.1.** *Let $U_i \in \mathbb{R}^{n_i \times d_i}$ be an orthonormal basis for $A_i \in \mathbb{R}^{n_i \times d_i}$. Then $U = \otimes_{i=1}^q U_i$ is an orthonormal basis for $A = \otimes_{i=1}^q A_i$.*

*Proof.* Note that the column norm of each column of $U$ is the product of column norms of the $U_i$'s, which are all 1. Thus $U$ has unit norm columns. It suffices then to show that all the singular values of $U$ are 1 or $-1$, but this follows from the fact that the singular values of $U$ are the product of singular values of the $U_i$'s, which completes the proof. ☐

**Corollary A.2.** *Let $A = \otimes_{i=1}^q A_i$, where $A_i \in \mathbb{R}^{n_i \times d_i}$. Fix any $\vec{i} = (i_1, \ldots, i_q) \in [n_1] \times [n_2] \times \cdots \times [n_q]$, and let $\vec{i}$ index into a row of $A$ in the natural way. Then the $\vec{i}$-th leverage score of $A$ is equal to $\prod_{j=1}^q \sigma_{i_j}(A_j)$, where $\sigma_t(B)$ is the t-th leverage score of a matrix $B$.*

*Proof.* Note $U = \otimes_{i=1}^q U_i$ is an orthonormal basis for $A = \otimes_{i=1}^q A_i$ by the prior Proposition. Now if $U_{\vec{i},*}$ is the $\vec{i}$-th row of $U$, then by fundamental properties of Kronecker products [VL00], we have $\|U_{\vec{i},*}\|_2 = \prod_{j=1}^q \|(U_j)_{i_j,*}\|_2$, which completes the proof. Note here that we used the fact that leverage scores are independent of the choice of orthonormal basis [Woo14]. ☐

**Proposition A.3** (Theorem 29 of [CW13]). *Given a matrix $A \in \mathbb{R}^{n \times d}$, let $\sigma \in \mathbb{R}^n$ be the $\ell_2$ leverage scores of $A$ (see definition 2.4). Then there is an algorithm which computes values $\widetilde{\sigma}_1, \widetilde{\sigma}_2, \ldots, \widetilde{\sigma}_n$ such that $\widetilde{\sigma}_i = (1 \pm \epsilon)\sigma_i$ simultaneously for all $i \in [n]$ with probability $1 - 1/n^c$ for any constant $c \geq 1$. The runtime is $\widetilde{O}(\mathrm{nnz}(A) + \mathrm{poly}(d/\epsilon))$.*

**Proposition A.4.** *Given $A = \otimes_{i=1}^q A_i$, where $A_i \in \mathbb{R}^{n_i \times d_i}$, there is an algorithm which, with probability $1 - 1/n^c$ for any constant $c \geq 1$, outputs a diagonal matrix $D \in \mathbb{R}^{n \times n}$ with $m$ non-zeros entries, such that $D_{i,i} = 1/(m\widetilde{\sigma}_i)$ is non-zero with probability $\widetilde{\sigma}_i \in (1 \pm 1/10)\sigma_i(A)$. The time required is $\widetilde{O}(\sum_{i=1}^q \mathrm{nnz}(A_i) + \mathrm{poly}(dq/\epsilon) + mq)$.*

*Proof.* By Proposition A.3, we can compute approximate leverage scores of each $A_i$ up to error $\Theta(1/q)$ in time $\widetilde{O}(\mathrm{nnz}(A_i) + \mathrm{poly}(d/\epsilon))$ with high probability. To sample a leverage score from $A$, it suffices to sample one leverage score from each of the $A_i$'s by Corollary A.2. The probability that a given row $\vec{i} = (i_1, \ldots, i_q) \in [n_1] \times [n_2] \times \cdots \times [n_q]$ of $A$ is chosen is $\prod_{j=1}^q \widetilde{\sigma}(A_j)_{i_j} = (1 \pm \Theta(1/q))^q \sigma_{\vec{i}}(A) = (1 \pm 1/10)\sigma_{\vec{i}}(A)$ as needed. Obtaining a sample takes $\widetilde{O}(1)$ time per $A_i$ (since a random number needs to be generated to $O(\log(n))$-bits of precision in expectation and with high probability to obtain this sample), thus $O(q)$ time overall, so repeating the sampling $M$ times gives the desired additive $mq$ runtime. ☐

The $q = 1$ version of the following result can be found in [CW13, SWZ19].

**Proposition A.5.** *Let $D \in \mathbb{R}^{n \times n}$ be the diagonal row sampling matrix generated via Proposition A.4, with $m = \Theta(1/(\delta\epsilon^2))$ non-zero entries. Let $A = \otimes_{i=1}^q A_i$ as above, and let $U \in \mathbb{R}^{n \times r}$ be an orthonormal basis for the column span of $A$, where $r = \mathrm{rank}(A)$. Then for any matrix $B$ with $n$ rows, we have*

$$\Pr\left[\|U^\top D^\top D B - U^\top B\|_F \leq \epsilon\|U\|_F\|B\|_F\right] \geq 1 - \delta$$

*Proof.* By definition of leverage scores and Proposition A.4, $D$ is a matrix which sample each row $U_{i,*}$ of $U$ with probability at least $(9/10)\|U_{i,*}\|_2/\|U\|_F$. Taking the average of $m$ such rows, we obtain the approximate matrix product result with error $O(1/\sqrt{\delta m})$ with probability $1 - \delta$ by Theorem 2.1 of [KV17]. ☐

502 We our now ready to prove the main theorem of this section, Theorem 3.1

503 **Theorem 3.1**(Kronecker product $\ell_2$ regression) *Let $D \in \mathbb{R}^{n \times n}$ be the diagonal row sampling matrix*
504 *generated via Proposition A.4, with $m = \Theta(1/(\delta\epsilon^2))$ non-zero entries, and let $A = \otimes_{i=1}^{q} A_i$,*
505 *where $A_i \in \mathbb{R}^{n_i \times d_i}$, and $b \in \mathbb{R}^n$, where $n = \prod_{i=1}^{q} n_i$ and $d = \prod_{i=1}^{q} d_i$. Then we have let*
506 *$\widehat{x} = \arg\min_{x \in \mathbb{R}^d} \|DAx - Db\|_2$, and let $x^* = \arg\min_{x' \in \mathbb{R}^d} \|Ax - b\|_2$. Then with probability*
507 *$1 - \delta$, we have*

$$\|A\widehat{x} - b\|_2 \leq (1 + \epsilon)\|Ax^* - b\|_2$$

508 *Moreover, the total runtime requires to compute $\widehat{x}$ is*

$$\widetilde{O}\left(\sum_{i=1}^{q} \mathrm{nnz}(A_i) + (dq/(\delta\epsilon))^{O(1)}\right).$$

509

510 *Proof.* Let $U$ be an orthonormal basis for the column span of $A$. By Lemma 3.3 of [CW09], we have
511 $\|A(\widehat{x} - x^*)\|_2 \leq 2\sqrt{\epsilon}\|Ax^* - b\|_2$. Note that while Lemma 3.3 of [CW09] uses a different sketching
512 matrix $D$ than us, the only property required for the proof of Lemma 3.3 is that $|U^\top D^\top DB - 
513 U^\top B\|_F \leq \sqrt{\epsilon/d}\|A\|_F\|B\|_F$ with probability at least $1 - \delta$ for any fixed matrix $B$, which we
514 obtain by Proposition A.5 by having $O(d/(\delta\epsilon^2))$ non-zeros on the diagonal of $D$). By the normal
515 equations, we have $A^\top(Ax^* - b) = 0$, thus $\langle A(\widehat{x} - x^*), (Ax^* - b)\rangle = 0$, and so by the Pythagorean
516 theorem we have

$$\|A\widehat{x} - b\|_2^2 = \|Ax^* - b\|_2^2 + \|A(\widehat{x} - x^*)\|_2^2 \leq (1 + 4\epsilon)\|Ax^* - b\|_2^2$$

517 Which completes the proof after rescaling of $\epsilon$. The runtime required to obtain the matrix $D$ is
518 $\widetilde{O}(\sum_{i=1}^{q} \mathrm{nnz}(A_i) + \mathrm{poly}(dq/\epsilon))$ by Proposition A.4, where we set $D$ to have $m = \Theta(d/(\delta\epsilon^2))$
519 non-zero entries on the diagonal. Once $D$ is obtained, one can compute $D(A + b)$ in time $O(md)$,
520 thus the required time is $O(\delta^{-1}(d/\epsilon)^2)$. Finally, computing $\widehat{x}$ once $DA, Db$ are computed requires
521 a single pseudo-inverse computation, which can be carried out in $O(\delta^{-1}d^3/\epsilon^2)$ time (since $DA$ now
522 has only $O(\delta^{-1}(d/\epsilon)^2)$ rows).

523 $\square$

## B  Missing Proofs from Section 3.1

525 We now give a complete proof of Theorem 3.2. Our high level approach follows that of [DDH+09].
526 Namely, we first obtain a vector $x'$ which is a $O(1)$ approximate solution to the optimal, and then
527 use the *residual error* $\rho \in \mathbb{R}^d$ of $x'$ to refine $x'$ to a $(1 \pm \epsilon)$ approximation $\widehat{x}$. The fact that $x'$ is
528 a constant factor approximation follows from our Lemma B.5. Given $x'$, by Lemma B.9 we can
529 efficiently compute the matrix $\Sigma$ which samples from the coordinates of the residual error $\rho = 
530 (A_1 \otimes \cdots \otimes A_q)x' - b$ in the desired runtime. The sampling lemma is the main technical lemma,
531 and requires a careful multi-part sketching and sampling routine. Given this $\Sigma$, the fact that $\widehat{x}$ is a
532 $(1 + \epsilon)$ approximate solution follows directly from Theorem 6 of [DDH+09]. Our main theorem
533 and its proof is stated below. The proof will utilize the lemmas and sampling algorithm developed
534 in the secitons which follow.

535 **Theorem 3.2** (Main result, $\ell_p$, $1 + \epsilon$-approximation). *Fix $1 \leq p < 2$. Then for any constant*
536 *$q = O(1)$, given matrices $A_1, A_2, \cdots, A_q$, where $A_i \in \mathbb{R}^{n_i \times d_i}$, let $n = \prod_{i=1}^{q} n_i$, $d = \prod_{i=1}^{q} d_i$. Let*
537 *$\widehat{x} \in \mathbb{R}^d$ be the output of Algorithm 2. Then*

$$\|(A_1 \otimes A_2 \otimes \cdots \otimes A_q)\widehat{x} - b\|_p \leq (1 + \epsilon) \min_{x \in \mathbb{R}^n} \|(A_1 \otimes A_2 \otimes \cdots \otimes A_q)x - b\|_p$$

538 *holds with probability at least $1 - \delta$. In addition, our algorithm takes*

$$\widetilde{O}\left(\left(\sum_{i=1}^{q} \mathrm{nnz}(A_i) + \mathrm{nnz}(b) + (d/\epsilon)^{O(1)}\right)\log(1/\delta)\right)$$

539 *time to output $\widehat{x} \in \mathbb{R}^d$.*

*Proof.* By Lemma B.5, the output $x'$ in line 16 of algorithm 3.1 is an 8 approximation of the optimal solution, and $x'$ is obtained in time $\widetilde{O}(\sum_{i=1}^q \mathrm{nnz}(A_i) + (dq/\epsilon)^{O(1)})$. We then obtain the residual error $\rho = (A_1 \otimes \cdots \otimes A_q)x' - b$ (implicitly). By Theorem 6 of [DDH$^+$09], if we let $\Sigma \in \mathbb{R}^{n \times n}$ be a row sampling matrix where $\Sigma_{i,i} = 1/\alpha_i^{1/p}$ with probability $\alpha_i = \min\{1, \max\{q_i, r_2 \frac{|\rho_i|^p}{\|\rho\|_p^p}\}$, where $q_i$ is the row sampling probability used in the sketch $\Pi$ from which $x'$ was obtained, and$r_2 = O(d^3/\epsilon^2 \log(1/\epsilon))$, then the solution to $\min_x \|\Sigma(A_1 \otimes \cdots \otimes A_q)x - \Sigma b\|_p$ will be a $(1 + \epsilon)$ approximately optimal solution. By Lemma B.9, we can obtain such a matrix $\Sigma$ in time $\widetilde{O}(\sum_{i=1}^q \mathrm{nnz}(A_i) + q\,\mathrm{nnz}(b) + (d\log(n)/(\epsilon\delta))^{O(q^2)})$, which completes the proof of correctness. Finally, note that we can solve the sketched regression problem $\min_x \|\Sigma(A_1 \otimes \cdots \otimes A_q)x - \Sigma b\|_p$ which has $O((d\log(n)/\epsilon)^{O(q^2)}(1/\delta))$ constraints and $d$ variables in time $O((d\log(n)/\epsilon)^{O(q^2)}(1/\delta))$ using linear programming for $p = 1$ (see [CLS19] for the state of the art linear program solver), or more generally interior point methods for convex programming for $p > 1$ (see [BCLL18] for the state of the art $\ell_p$ solver).

Now to boost the failure probability from a $O(1/\delta)$ to $\log(1/\delta)$ dependency, we do the following. We run the above algorithm with $\delta = 1/10$, so that our output $\widehat{x} \in \mathbb{R}^d$ is a $(1 + \epsilon)$ approximation with probability $9/10$. Now note that we actually have an $(1 + \epsilon)$ estimate of the cost $\|(A_1 \otimes A_2 \otimes \cdots \otimes A_q)\widehat{x} - b\|_p$ of the solution $\widehat{x}$, which is simply given by $\|\Sigma(A_1 \otimes A_2 \otimes \cdots \otimes A_q)\widehat{x} - \Sigma b\|_p$ where $\Sigma$ is the sampling matrix used to compute $\widehat{x}$. Thus we can simply repeat the above process $O(\log(1/\delta))$ times, and take the solution with the minimal cost overall. $\square$

We start by defining a tensor operation which will be useful for our analysis.

**Definition B.1** ( $((\cdot, \ldots, \cdot), \cdot)$ operator for tensors and matrices). *Given tensor $A \in \mathbb{R}^{d_1 \times d_2 \times \cdots \times d_q}$ and matrices $B_i \in \mathbb{R}^{n_i \times d_i}$ for $i \in [q]$, we define the tensor $((B_1, B_2, \ldots, B_q), A) \in \mathbb{R}^{n_1 \times n_2 \times \cdots \times n_q}$:*

$$((B_1, B_2, \ldots, B_q), A)_{i_1, \ldots, i_q} = \sum_{i_1'=1}^{d_1} \sum_{i_2'=1}^{d_2} \cdots \sum_{i_q'=1}^{d_q} A_{i_1', i_2', \ldots, i_q'} \prod_{\ell=1}^{q} (B_\ell)_{i_\ell, i_\ell'}$$

*Observe for the case of $q = 2$, we just have $((B_1, B_2), A) = B_1 A B_2^\top \in \mathbb{R}^{n_1 \times n_2}$.*

Using the above notation, we first prove a result about reshaping tensors.

**Lemma B.2** (Reshaping). *Given matrices $A_1, A_2, \cdots, A_q \in \mathbb{R}^{n_i \times d_i}$ and a tensor $B \in \mathbb{R}^{n_1 \times n_2 \times \cdots \times n_q}$, let $n = \prod_{i=1}^q n_i$ and let $d = \prod_{i=1}^d d_i$. Let $b$ denote the vectorization of $B$. For any tensor $X \in \mathbb{R}^{d_1 \times d_2 \times \cdots \times d_q}$, we have $\|((A_1, A_2, \cdots, A_q), X) - B\|_\xi$ is equal to $\|(A_1 \otimes A_2 \otimes \cdots \otimes A_q)x - b\|_\xi$ where $\xi$ is any entry-wise norm (such as an $\ell_p$-norm) and $x$ is the vectorization of $X$. See Definition B.1 of the $((\cdot, \ldots, \cdot), \cdot)$ tensor operator.*

*Observe, for the case of $q = 2$, this is equivalent to the statement that $\|A_1 X A_2^\top - B\|_\xi = \|(A_1 \otimes A_2)x - b\|_\xi$.*

*Proof.* For the pair $x \in \mathbb{R}^d$, $X \in \mathbb{R}^{d_1 \times d_2 \times \cdots \times d_q}$, the connection is the following: $\forall i_1 \in [d_1], \ldots, i_q \in [d_q]$,

$$x_{i_1 + \sum_{l=2}^q (i_l - 1) \cdot \prod_{t=1}^{l-1} d_t} = X_{i_1, \cdots, i_q}.$$

Similarly, for $b \in \mathbb{R}^n$, $B \in \mathbb{R}^{n_1 \times n_2 \times \cdots \times n_q}$, for any $j_1, \in [n_1], \ldots, j_q \in [n_q]$,

$$b_{j_1 + \sum_{l=2}^q (j_l - 1) \cdot \prod_{t=1}^{l-1} n_t} = B_{j_1, j_2, \cdots, j_q}.$$

For simplicity, for any $(i_1, \ldots, i_q) \in [d_1] \times \cdots \times [d_q]$ and $(j_1, \ldots, j_q) \in [n_1] \times \cdots \times [n_q]$ we define $\vec{i} = i_1 + \sum_{l=2}^q (i_l - 1) \cdot \prod_{t=1}^{l-1} d_t$ and similarly $\vec{j} = j_1 + \sum_{l=2}^q (j_l - 1) \cdot \prod_{t=1}^{l-1} n_t$. Then we can simplify the above relation and write $x_{\vec{i}} = X_{i_1, i_2, \cdots, i_q}$, and $b_{\vec{j}} = B_{j_1, j_2, \cdots, j_q}$.

578     For a matrix $Z$, let $Z_{i,*}$ denote the $i$-th row of $Z$. We consider the $\vec{j}$-th entry of $(A_1 \otimes A_2 \otimes \cdots \otimes A_q)x$,

$$((A_1 \otimes A_2 \otimes \cdots \otimes A_q)x)_{\vec{j}} = \left\langle (A_1 \otimes A_2 \otimes \cdots \otimes A_q)_{\vec{j},*} \cdot x \right\rangle$$

$$= \sum_{i_1=1}^{d_1} \sum_{i_2=1}^{d_2} \cdots \sum_{i_q=1}^{d_q} \left( \prod_{l=1}^{q} (A_l)_{j_l,i_l} \right) \cdot x_{\vec{i}}$$

$$= \sum_{i_1=1}^{d_1} \sum_{i_2=1}^{d_2} \cdots \sum_{i_q=1}^{d_q} \left( \prod_{l=1}^{q} (A_l)_{j_l,i_l} \right) \cdot X_{i_1,i_2,\cdots,i_q}$$

$$= ((A_1, A_2, \cdots, A_q), X)_{j_1,\ldots,j_q}.$$

579     Where the last equality is by Definition (B.1). Since we also have $b_{\vec{j}} = B_{j_1,\ldots,j_q}$, this completes the
580     proof of the Lemma. $\qquad\qquad\qquad\qquad\qquad\qquad\qquad\qquad\qquad\qquad\qquad\qquad\qquad\qquad\qquad\square$

## B.1    Sampling from an $\ell_p$-Well-Conditioned Base

582     In this Section, we discuss the first half of Algorithm 2 which computes $x' \in \mathbb{R}^d$, which we will
583     show is a $O(1)$-approximate solution to the optimal. First note that by Lemma 2.3 together with fact
584     2.6, we know that $A_i R_i^{-1}$ is an $\ell_p$ well conditioned basis for $A_i$ (recall this means that $A_i R_i^{-1}$ is
585     a $(\alpha, \beta, p)$ well conditioned basis for $A$, and $\beta/\alpha = d_i^{O(1)}$) with probability $1 - O(1/q)$, and we
586     can then union bound over this occurring for all $i \in [q]$. Given this, we now prove that $(A_1 R_1^{-1} \otimes$
587     $A_2 R_2^{-1} \otimes \cdots \otimes A_q R_q^{-1})$ is a well conditioned basis for $(A_1 \otimes A_2 \otimes \cdots \otimes A_q)$.

588     **Lemma B.3.** *Let $A_i \in \mathbb{R}^{n_i \times d_i}$ and $R_i \in \mathbb{R}^{d_i \times d_i}$. Then if $A_i R_i^{-1}$ is a $(\alpha_i, \beta_i, p)$ well-conditioned*
589     *basis for $A_i$ for $i = 1, 2, \ldots, q$, we have for all $x \in \mathbb{R}^{d_1 \cdots d_q}$:*

$$\prod_{i=1}^{q} \alpha_i \|x\|_p \leq \|(A_1 R_1^{-1} \otimes A_2 R_2^{-1} \otimes \cdots \otimes A_q R_q^{-1})x\|_p \leq \prod_{i=1}^{q} \beta_i \|x\|_p$$

590     *Proof.* We first consider the case of $q = 2$. We would like to prove

$$\alpha_1 \alpha_2 \|x\|_p \leq \|(A_1 R_1^{-1} \otimes A_2 R_2^{-1})x\|_p \leq \beta_1 \beta_2 \|x\|_p,$$

591     First note, by the reshaping Lemma B.2, this is equivalent to

$$\alpha_1 \alpha_2 \|X\|_p \leq \|A_1 R_1^{-1} X (R_2^{-1} A_2)^\top\|_p \leq \beta_1 \beta_2 \|X\|_p.$$

592     Where $X \in \mathbb{R}^{d_1 \times d_2}$ is the tensorization of $x$. We first prove one direction. Let $U_1 = A_1 R_1^{-1}$ and
593     $U_2 = A_2 R_2^{-1}$. We have

$$\|U_1 X U_2^\top\|_p^p = \sum_{i_2=1}^{n_2} \|U_1 (X U_2^\top)_{i_2}\|_p^p$$

$$\leq \sum_{i_2=1}^{n_2} \beta_1^p \|(X U_2^\top)_{i_2}\|_p^p$$

$$= \beta_1^p \|X U_2^\top\|_p^p$$

$$\leq \beta_1^p \beta_2^p \|X\|_p^p,$$

594     where the first step follows from rearranging, the second step follows from the well-conditioned
595     property of $U_1$, the third step follows from rearranging again, the last step follows from the well-
596     conditioned property of $U_2$. Similarly, we have

$$\|U_1 X U_2^\top\|_p^p = \sum_{i_2=1}^{n_2} \|U_1 (X U_2^\top)_{i_2}\|_p^p$$

$$\geq \sum_{i_2=1}^{n_2} \alpha_1^p \|(X U_2^\top)_{i_2}\|_p^p$$

$$= \alpha_1^p \|X U_2^\top\|_p^p$$

$$\geq \alpha_1^p \alpha_2^p \|X\|_p^p,$$

597 where again the first step follows from rearranging, the second step follows from the well-
598 conditioned property of $U_1$, the third step follows from rearranging again, the last step follows
599 from the well-conditioned property of $U_2$.

600 In general, for arbitrary $q \geq 2$, similarly using our reshaping lemma, we have

$$\|(\otimes_{i=1}^q (A_i R_i^{-1}))x\|_p \geq \prod_{i=1}^q \alpha_i \|x\|_p,$$

$$\|(\otimes_{i=1}^q (A_i R_i^{-1}))x\|_p \leq \prod_{i=1}^q \beta_i \|x\|_p.$$

601 □

602 Putting this together with fact 2.6, and noting $d = d_1 \cdots d_q$, we have

603 **Corollary B.4.** *Let $A_i R_i^{-1}$ be as in algorithm 2. Then we have for all $x \in \mathbb{R}^{d_1 \cdots d_q}$:*

$$(1/d)^{O(1)} \|x\|_p \leq \|(A_1 R_1^{-1} \otimes \cdots \otimes A_q R_q^{-1})x\|_p \leq d^{O(1)} \|x\|_p,$$

604 *In other words, $(A_1 R_1^{-1} \otimes \cdots \otimes A_q R_q^{-1})$ is a well conditioned $\ell_p$ basis for $(A_1 \otimes \cdots \otimes A_q)$*

605 From this, we can obtain the following result.

**Lemma B.5.** *Let $x' \in \mathbb{R}^d$ be the output of the $O(1)$-Approximate $\ell_p$ Regression Procedure in Algorithm 2. Then with probability $99/100$ we have*

$$\|(A_1 \otimes \cdots \otimes A_q)x' - b\|_p \leq 8 \min_x \|(A_1 \otimes \cdots \otimes A_q)x - b\|_p$$

606 *Moreover, the time required to compute $x'$ is $\widetilde{O}(\sum_{i=1}^q \text{nnz}(A_i) + (dq/\epsilon)^{O(1)})$.*

607 *Proof.* By Theorem 6 of [DDH+09], if we let $\Pi$ be a diagonal row sampling matrix such that
608 $\Pi_{i,i} = 1/q_i^{1/p}$ with probability $q_i \geq \min\{1, r_1 \frac{\|U_{i,*}\|_p^p}{\|U\|_p^p}\}$, where $U$ is a $\ell_p$ well-conditioned basis for
609 $(A_1 \otimes \cdots \otimes A_q)$ and $r_1 = O(d^3)$, then the solution $x'$ to

$$\min_x \|\Pi((A_1 \otimes \cdots \otimes A_q)x - b)\|$$

610 will be a 8-approximation. Note that we can solve the sketched regression problem $\min_x \|\Pi((A_1 \otimes
611 \cdots \otimes A_q)x' - b\|$ which has $O(\text{poly}(d/\epsilon))$ constraints and $d$ variables in time $\text{poly}(d/\epsilon)$ using
612 linear programming for $p = 1$ (see [CLS19] for the state of the art linear program solver), or more
613 generally interior point methods for convex programming for $p > 1$ (see [BCLL18] for the state of
614 the art $\ell_p$ solver).

615 Then by Corollary B.4, we know that setting $U = (A_1 R_1^{-1} \otimes \cdots \otimes A_q R_q^{-1})$ suffices, so now we
616 must sample rows of $U$. To do this, we must approximately compute the norms of the rows of $U$.
617 Here, we use the fact that $\|\cdot\|_p^p$ norm of a row of $(A_1 R_1^{-1} \otimes \cdots \otimes A_q R_q^{-1})$ is the product of the row
618 norms of the $A_i R_i^{-1}$ that correspond to that row. Thus it suffices to sample a row $j_i$ from each of
619 the $A_i R_i^{-1}$'s with probability at least $\min\{1, r_1 \|(A_i R_i^{-1})_{j_i,*}\|_p^p / \|A_i R_i^{-1}\|_p^p\}$ for each $i \in [q]$.

620 To do this, we must estimate all the row norms $\|(A_i R_i^{-1})_{j_i,*}\|_p^p$ to $(1 \pm 1/10)$ error. This is done
621 in steps $7 - 10$ of Algorithm 2, which uses dense $p$-stable sketches $Z \in \mathbb{R}^{d \times \tau}$, and computes
622 $(A_i R_i^{-1} Z)$, where $\tau = \Theta(\log(n))$. Note that computing $R_i^{-1} Z \in \mathbb{R}^{d \times \tau}$ requires $\widetilde{O}(d^2)$. Once
623 computed, $A_i(R_i^{-1}Z)$ can be computed in $\widetilde{O}(\text{nnz}(A_i))$ time. We then take the median of the coor-
624 dinates of $(A_i R_i^{-1} Z)$ (normalized by the median of the $p$-stable distribution $\mathcal{D}_p$, which can be effi-
625 ciently approximated to $(1 \pm \epsilon)$ in $O(\text{poly}(1/\epsilon))$ time, see Appendix A.2 of [KNW10] for details)
626 as our estimates for the row norms. This is simply the Indyk median estimator [Ind06], and gives
627 a $(1 \pm 1/10)$ estimate $a_{i,j}$ of all the row norms $\|(A_i R_i^{-1})_{j,*}\|_p^p$ with probability $1 - 1/\text{poly}(n)$.
628 Then it follows by Theorem 6 of [DDH+09] that $x'$ is a 8-approximation of the optimal solution
629 with probability $99/100$ (note that we amplified the probability by increasing the sketch sizes $S_i$ by
630 a constant factor), which completes the proof.

631 □

## B.2 $\ell_p$ Sampling From the residual of a $O(1)$-factor approximation

By Lemma B.5 in the prior section, we know that the $x'$ first returned by the in algorithm 2 is a 8-approximation. We now demonstrate how we can use this $O(1)$ approximation to obtain a $(1 + \epsilon)$ approximation. The approach is again to sample rows of $(A_1 \otimes \cdots \otimes A_q)$. But instead of sampling rows with the well-conditioned leverage scores $q_i$, we now sample the $i$-th row with probability $\alpha_i = \min\{1, \max\{q_i, r_2|\rho_i|^p/\|\rho\|_p^p\}\}$, where $\rho = (A_1 \otimes \cdots \otimes A_q)x' - b \in \mathbb{R}^n$ is the *residual error* of the $O(1)$-approximation $x'$. Thus we must now determine how to sample quickly from the residuals $|\rho_i|^p/\|\rho\|_p^p$. Our sampling algorithm will need a tool originally developed in the streaming literature.

**Count-sketch for heavy hitters with the Dyadic Trick.** We now introduce a sketch $S$ which finds the $\ell_2$ heavy hitters in a vector $x$ efficently. This sketch $S$ is known as count-sketch for heavy hitters with the Dyadic Trick. To build $S$ we first stack $\Theta(\log(n))$ copies of the *count sketch matrix* $S^i \in \mathbb{R}^{k' \times n}$ [CW13]. The matrix $S^i$ is constructed as follows. $S^i$ has exactly one non-zero entry per column, which is placed in a uniformly random row, and given the value 1 or $-1$ uniformly at random. For $S^i$, let $h_i : [n] \to [k']$ be such that $h_i(t)$ is the row with the non-zero entry in the $t$-th column of $S^i$, and let $g_i : [n] \to \{1, -1\}$ be such that the value of that non-zero entry is $g_i(t)$. Note that the $h_i, g_i$ can be implemented as 4-wise independent hash functions. Fix any $x \in \mathbb{R}^n$. Then given $S^1x, S^2x, \cdots, S^{\Theta(\log(n))}x$, we can estimate the value of any coordinate $x_j$ by $\text{median}_{i \in \Theta \log(n)}\{g_i(j)(S^ix)_{h_i(j)}\}$.

It is well-known that this gives an estimate of $x_j$ with additive error $\Theta(1/\sqrt{k'})\|x\|_2$ with probability $1 - 1/\text{poly}(n)$ for all $j \in [n]$ [CCFC04]. However, naively, to find the heaviest coordinates in $x$, that is all coordinates $x_j$ with $|x_j| \geq \Theta(1/\sqrt{k'})\|x\|_2$, one would need to query $O(n)$ estimates. This is where the Dyadic trick comes in [CM05]. We repeat the above process $\Theta(\log(n))$ times, with matrices $S^{(i,j)}$, for $i, j \in \Theta(\log(n))$. Importantly, however, in $S^{(i,j)}$, for all $t, t' \in [n]$ such that the first $j$ most significant bits in their binary identity representation are the same, we set $h_{(i,j)}(t) = h_{(i,j)}(t')$, effectively collapsing these identities to one. To find a heavy item, we can then query the values of the *two* identities from $S^{(1,1)}, S^{(2,1)}, \cdots, S^{(\Theta(\log(n)),1)}$, and recurse into all the portions which have size at least $\Theta(1/\sqrt{k'})\|x\|_2$. It is easy to see that we recurse into at most $O(k')$ such pieces in each of the $\Theta(\log(n))$ levels, and it takes $O(\log(n))$ time to query a single estimate, from which the desired runtime of $O(k' \log^2(n))$ is obtained. For a further improvement on size $k$ of the overall sketched required to quickly compute $Q$, see [LNNT16]. We summarize this construction below in definition B.6.

**Definition B.6** (Count-sketch for heavy hitters with Dyadic Trick [CCFC04, LNNT16])**.** *There is a randomized sketch $S \in \mathbb{R}^{k \times n}$ with $k = O(\log^2(n)/\epsilon^2)$ such that, for a fixed vector $x \in \mathbb{R}^n$, given $Sx \in \mathbb{R}^k$, one can compute a set $Q \subset [n]$ with $|Q| = O(1/\epsilon^2)$ such that $\{i \in [n] \mid |x_i| \geq \epsilon\|x\|_2\} \subseteq Q$ with probability $1 - 1/\text{poly}(n)$. Moreover, $Sx$ can be computed in $O(\log^2(n)\,\text{nnz}(x))$ time. Given $Sx$, the set $Q$ can be computed in time $O(k)$.*

We begin with some notation. For a vector $y \in \mathbb{R}^n$, where $n = n_1 \cdots n_q$, one can index any entry of $y_i$ via $\vec{i} = (i_1, i_2, \cdots, i_q) \in [n_1] \times \cdots \times [n_q]$ via $i = i_1 + \sum_{j=2}^q (i_j - 1) \prod_{l=1}^{i_j - 1} n_l$. It will useful to index into such a vector $y$ interchangably via a vector $y_{\vec{i}}$ and an index $y_j$ with $j \in [n]$. For any set of subsets $T_i \subset [n_i]$, we can define $y_{T_1 \times \cdots T_q} \in \mathbb{R}^n$ as $y$ restricted to the $\vec{i} \in T_1 \times \cdots \times T_q$. Here, by restricted, we mean the coordinates in $y$ that are not in this set are set equal to 0. Similarly, for a $y \in \mathbb{R}^{n_i}$ and $S \subset [n_i]$, we can define $y_S$ as $y$ restricted to the coordinates in $S$. Note that in Algorithm 4, $\mathbb{I}_n$ denotes the $n \times n$ identity matrix for any integer $n$. We first prove a proposition on the behavior of Kronecker products of $p$-stable vectors, which we will need in our analysis.

**Proposition B.7.** *Let $Z_1, Z_2, \cdots, Z_q$ be independent vectors with entries drawn i.i.d. from the $p$-stable distribution, with $Z_i \in \mathbb{R}^{n_i}$. Now fix any $i \in [q]$, and any $x \in \mathbb{R}^n$, where $n = n_1 n_2 \cdots n_q$. Let $e_j \in \mathbb{R}^{n_i}$ be the $j$-th standard basis column vector for any $j \in [n_i]$. Let $\Gamma(i, j) = [n_1] \times [n_2] \times \cdots \times [n_{i-1}] \times \{j\} \times [n_{i+1}] \times \cdots \times [n_q]$. Define the random variable*

$$\mathcal{X}_{i,j}(x) = |(Z_1 \otimes Z_1 \otimes \cdots \otimes Z_{i-1} \otimes e_j^\top \otimes Z_{i+1} \otimes \cdots \otimes Z_q)x|^p.$$

*Then for any $\lambda > 1$, with probability at least $1 - O(q/\lambda)$ we have*

$$\|x_{\Gamma(i,j)}\|_p^p/\lambda^q \leq \mathcal{X}_{i,j}(x) \leq (\lambda \log(n))^q \|x_{\Gamma(i,j)}\|_p^p$$

---

**Algorithm 4** Algorithm to $\ell_p$ sample $\Theta(r_2)$ entires of $\rho = (A_1 \otimes \cdots \otimes A_q)x' - b$

---

1: **procedure** RESIDUAL $\ell_p$ SAMPLE($\rho, r_2$)
2:     $r_3 \leftarrow \Theta(r_2 \log^{q^2}(n)/\delta)$.
3:     Generate i.i.d. $p$-stable vectors $Z^{1,j}, Z^{2,j}, \ldots, Z^{q,j} \in \mathbb{R}^n$ for $j \in [\tau]$ for $\tau = \Theta(\log(n))$
4:     $T \leftarrow \emptyset$                                    $\triangleright$ sample set to return
5:     Pre-compute and store $Z^{i,j} A_i \in \mathbb{R}^{1 \times d_i}$ for all $i \in [q]$ and $j \in [\tau]$
6:     Generate count-sketches for heavy hitters $S^i \in \mathbb{R}^{k \times n_i}$ of Definition B.6 for all $i \in [q]$, where $k = O(\log^2(n)r_3^{O(1)})$.
7:     **for** $t = 1, 2, \ldots, r_3$ **do**
8:         $s = (s_1, \ldots, s_q) \leftarrow (\emptyset, \ldots, \emptyset)$                 $\triangleright$ next sample to return
9:         $w^j \leftarrow \left( (\mathbb{I}_{n_1}) \otimes (\bigotimes_{k=2}^q Z^{k,j}) \rho \right) \in \mathbb{R}^{n_1}$        $\triangleright \mathbb{I}_n \in \mathbb{R}^{n \times n}$ is identity
10:         Define $w \in \mathbb{R}^{n_1}$ by $w_l = \text{median}_{j \in [\tau]}\{|w_l^j|\}$ for $l \in [n_1]$
11:         Sample $j^* \in [n_1]$ from the distribution $\left( \frac{|w_1|^p}{\|w\|_p^p}, \frac{|w_2|^p}{\|w\|_p^p}, \ldots, \frac{|w_{n_1}|^p}{\|w\|_p^p} \right)$
12:         $s_1 \leftarrow j^*$
13:         **for** $i = 2, \ldots, q$ **do**
14:             **for** $j \in [\tau]$ **do**
15:                 Write $e_{a_k}^\top \in \mathbb{R}^{1 \times n_k}$ as the standard basis vector
16:                 $v_i^j \leftarrow S^i \left( (\bigotimes_{k=1}^{i-1} e_{a_k}^\top) \otimes (\mathbb{I}_{n_i}) \otimes (\bigotimes_{k=i+1}^q Z^{k,j}) \rho \right) \in \mathbb{R}^k$
17:                 Compute heavy hitters $H_{i,j} \subset [n_i]$ from $v_i^j$      $\triangleright$ Definition B.6
18:                 $\beta_i^j \leftarrow \left( (\bigotimes_{k=1}^{i-1} e_{a_k}^\top) \otimes (\bigotimes_{k=i}^q Z^{k,j}) \rho \right) \in \mathbb{R}$
19:             **end for**
20:             Define $\beta_i \in \mathbb{R}^{k'}$ by $\beta_i = \text{median}_{j \in [\tau]}\{|\beta_i^j|^p\}$
21:             $H_i = \cup_{j=1}^\tau H_{i,j}$
22:             $\gamma_i \leftarrow \text{median}_{j \in [\tau]} \left( (\bigotimes_{k=1}^{i-1} e_{a_k}^\top) \otimes Z_{[n_i] \setminus H_i}^{i,j} \otimes (\bigotimes_{k=i+1}^q Z^{k,j}) \rho \right) \in \mathbb{R}$
23:             **if** with probability $1 - \gamma_i/\beta_i$ **then**
24:                 Draw $\xi \in H_i$ with probability

$$\frac{\text{median}_{j \in \tau} \left| \left( (\bigotimes_{k=1}^{i-1} e_{a_k}^\top) \otimes (e_\xi^\top) \otimes (\bigotimes_{k=i+1}^q Z^{k,j}) \rho \right) \right|^p}{\sum_{\xi' \in H_i} \text{median}_{j \in \tau} \left| \left( (\bigotimes_{k=1}^{i-1} e_{a_k}^\top) \otimes (e_{\xi'}^\top) \otimes (\bigotimes_{k=i+1}^q Z^{k,j}) \rho \right) \right|^p}$$

25:                 $s_i \leftarrow \xi$
26:             **else**                         $\triangleright s_i$ was not sampled as a heavy hitter
27:                 Randomly partition $[n_i]$ into $\Omega_1^i, \Omega_2^i, \ldots, \Omega_\eta^i$ with $\eta = \Theta(r_3^2)$
28:                 Sample $t \sim [\eta]$ uniformly at random
29:                 **for** $j \in \Omega_t \setminus H_i$ **do**
30:                     $\theta_j = \text{median}_{l \in [\tau]} \left( |(\bigotimes_{k=1}^{i-1} e_{a_k}^\top) \otimes (e_j^\top) \otimes (\bigotimes_{k=i+1}^q Z^{k,l}) \rho|^p \right)$
31:                 **end for**
32:                 Sample $s_i \leftarrow j^*$ from the distribution $\{ \frac{\theta_j}{\sum_{j' \in \Omega_t \setminus H_i} \theta_{j'}} \}_{j \in \Omega_t \setminus H_i}$
33:             **end if**
34:         **end for**
35:         $T \leftarrow S \cup s$ where $s = (s_1, \ldots, s_q)$
36:     **end for**
37:     **return** sample set $T$
38: **end procedure**

---

*Proof.* First observe that we can reshape $y = x_\Gamma \in \mathbb{R}^m$ where $m = n/n_i$, and re-write this random variable as $\mathcal{X}_{i,j}(x) = |(Z_1 \otimes Z_2 \otimes \cdots \otimes Z_{q-1})y|^p$. By reshaping Lemma B.2, we can write this as $|(Z_1 \otimes Z_2 \otimes \cdots \otimes Z_{q-2})Y Z_{q-1}^\top|^p$, where $Y \in \mathbb{R}^{m/n_{q-1} \times n_{q-1}}$. We first prove a claim. In the following, for a matrix $A$, let $\|A\|_p^p = \sum_{i,j} |A_{i,j}|^p$.

**Claim B.8.** *Let $Z$ be any p-stable vector and $X$ a matrix. Then for any $\lambda > 1$, with probability $1 - O(1/\lambda)$, we have*

$$\lambda^{-1}\|X\|_p^p \le \|XZ\|_p^p \le \log(n)\lambda\|X\|_p^p.$$

*Proof.* By $p$-stability, each entry of $|(XZ)_i|^p$ is distributed as $|z_i|^p\|X_{i,*}\|_p^p$, where $z_i$ is again $p$-stable (but the $z_i's$ are not independent). Now $p$-stables have tails that decay at the rate $\Theta(1/x^p)$ (see Chapter 1.5 of [Nol07]), thus $\Pr[|z_i|^p > x] = O(1/x)$ for any $x > 0$. We can condition on the fact that $z_i < \lambda \cdot n^{10}$ for all $i$, which occurs with probability at least $1 - n^{-9}/\lambda$ by a union bound. Conditioned on this, we have $\mathbb{E}[|z_i|^p] = O(\log(n))$ (this can be seen by integrating over the truncated tail $O(1/x)$), and the upper bound then follows from a application of Markov's inequality.

For the lower bound Let $Y_i$ be an indicator random variable indicating the event that $|z_i|^p < 2/\lambda$. Now $p$-stables are anti-concentrated, namely, their pdf is upper bounded by a constant everywhere. It follows that $\Pr[Y_i] < c/\lambda$ for some constant $c$. By Markov's inequality $\Pr[\sum_i Y_i\|X_{i,*}\|_p^p > \|X\|_p^p/2] < O(1/\lambda)$. Conditioned on this, the remaining $\|X\|_p^p/2$ of the $\ell_p$ mass shrinks by less than a $2/\lambda$ factor, thus $\|XZ\|_p^p > (\|X\|_p^p/2)(2/\lambda) = \|X\|_p^p/\lambda$ as needed. $\square$

By the above claim, we have $\|Y\|_p/\lambda^{1/p} \le \|YZ_{q-1}^\top\|_p \le (\log(n)\lambda)^{1/p}\|Y\|_p$ with probability $1 - O(1/\lambda)$. Given this, we have $\mathcal{X}_{i,j}(x) = |(Z_1 \otimes Z_2 \otimes \cdots \otimes Z_{q-2})y'|^p$, where $\|Y\|_p/\lambda^{1/p} \le \|y'\|_p \le (\log(n)\lambda)^{1/p}\|Y\|_p$. We can inductively apply the above argument, each time getting a blow up of $(\log(n)\lambda)^{1/p}$ in the upper bound and $(1/\lambda)^p$ in the lower bound, and a failure probability of $(1/\lambda)$. Union bounding over all $q$ steps of the induction, the proposition follows.

$\square$

**Lemma B.9.** *Fix any $r_2 \ge 1$, and suppose that $x' = \min_x \|\Pi(A_1 \otimes \cdots \otimes A_q)x - \Pi b\|_p$ and $\Pi \in \mathbb{R}^{n \times n}$ is a row sampling matrix such that $\Pi_{i,i} = 1/q_i^{1/p}$ with probability $q_i$. Define the residual error $\rho = (A_1 \otimes \cdots \otimes A_q)x' - b \in \mathbb{R}^n$. Then Algorithm 4, with probability $1 - \delta$, succeeds in outputting a row sampling matrix $\Sigma \in \mathbb{R}^{n \times n}$ such that $\Sigma_{i,i} = 1/\alpha_i^{1/p}$ with probability $\alpha_i = \min\{1, \max\{q_i, r_3|\rho_i|^p/\|\rho\|_p^p\}\}$ for some $r_3 \ge r_2$, and otherwise $\Sigma_{i,i} = 0$. The algorithm runs in time*

$$\widetilde{O}\left(\sum_{i=1}^{q} \mathrm{nnz}(A_i) + q\,\mathrm{nnz}(b) + (r_2\log(n)/\delta)^{O(q^2)}\right).$$

*Proof.* The algorithm is given formally in figure 4. We analyze the runtime and correctness here.

**Proof of Correctness.** The approach of the sampling algorithm is as follows. Recall that we can index into the coordinates of $\rho \in \mathbb{R}^n$ via $\vec{a} = (a_1, \ldots, a_q)$ where $a_i \in [n_i]$. We build the coordinates of $\vec{a}$ one by one. To sample a $\vec{a} \in \prod_{i=1}^q [n_i]$, we can first sample $a_1 \in [n_1]$ from the distribution $\Pr[a_1 = j] = \sum_{\vec{u}:u_1=j}|\rho_{\vec{u}}|^p/(\sum_{\vec{u}}|\rho_{\vec{u}}|^p)$. Once we fix $a_1$, we can sample $a_2$ from the conditional distribution distribution $\Pr[a_2 = j] = \sum_{\vec{u}:u_2=j,u_1=a_1}|\rho_{\vec{u}}|^p/(\sum_{\vec{u}:u_1=a_1}|\rho_{\vec{u}}|^p)$, and so on. For notation, given a vector $\vec{a} = (a_1, \ldots, a_{i-1})$, let $\Delta(\vec{a}) = \{\vec{u} \in [n_1] \times \cdots \times [n_q] \mid a_j = y_j \text{ for all } j = 1, 2, \ldots, i-1\}$. Then in general, when we have sampled $\vec{a} = (a_1, \ldots, a_{i-1})$ for some $i \le q$, we need to sample $a_i \leftarrow j \in [n_k]$ with probability

$$\Pr[a_i = j] = \sum_{\vec{u} \in \Delta(\vec{a}):u_i=j} |\rho_{\vec{u}}|^p / \left(\sum_{\vec{u} \in \Delta(\vec{a})} |\rho_{\vec{u}}|^p\right).$$

We repeat this process to obtain the desired samples. Note that to sample efficiently, we will have to compute these aforementioned sampling probabilities approximately. Because of the error in approximating, instead of returning $r_2$ samples, we over-sample and return $r_3 = \Theta(r_2\log^{q^2}(n))$ samples.

The first step is of the algorithm is to generate the $p$-stable vectors $Z^{i,j} \in \mathbb{R}^{n_i}$ for $i \in [q]$ and $j = 1, 2, \ldots, \Theta(\log(n))$. We can pre-compute and store $Z^{i,j}A_i$ for $i \in [q]$, which takes

726    $\widetilde{O}(\sum_{i=1}^{q} \mathrm{nnz}(A_i))$ time. We set $w^j \leftarrow \left((\mathbb{I}_{n_1}) \otimes (\bigotimes_{k=2}^{q} Z^{k,j})\rho\right) \in \mathbb{R}^{n_1}$ and define $w \in \mathbb{R}^{n_1}$

727    by $w_l = \mathrm{median}_{j \in [\tau]}\{|w_l^j|\}$ for $l \in [n_1]$. Observe that $w_l^j$ is an estimate of $\sum_{\vec{u}:u_1=l} |\rho_{\vec{u}}|^p$. By

728    Proposition B.7, it is a $(c \log(n))^q$ approximation with probability at least $3/4$ for some constant $c$.

729    Taking the median of $\Theta(\log(n))$ repetitions, we have that

$$c^{-q} \cdot \sum_{\vec{u}:u_1=l} |\rho_{\vec{u}}|^p \leq |w_l|^p \leq (c \log(n))^q \cdot \sum_{\vec{u}:u_1=l} |\rho_{\vec{u}}|^p$$

730    with probability $1 - 1/\mathrm{poly}(n)$, and we can then union bound over all such estimates every con-

731    ducted over the course of the algorithm. We call the above estimate $|w_l|^p$ a $O((c \log(n))^q)$-error

732    estimate of $\sum_{\vec{u}:u_1=l} |\rho_{\vec{u}}|^p$. Given this, we can correctly and independently sample the first coor-

733    dinate of each of the $\Theta(r_3)$ samples. We now describe how to sample the $i$-th coordinate. So

734    in general, suppose we have sampled $(a_1, ..., a_{i-1})$ so far, and we need to now sample $a_i \in [n_i]$

735    conditioned on $(a_1, ..., a_{i-1})$. We first consider

$$W^{i,k} = \left((\bigotimes_{k=1}^{i-1} e_{a_k}^{\top}) \otimes (\mathbb{I}_{n_i}) \otimes (\bigotimes_{k=i+1}^{q} Z^{k,j})\rho\right) \in \mathbb{R}^{n_i}$$

736    Note that the $j$-th coordinate $W_j^{i,k}$ for $W^{i,k}$ is an estimate of $\sum_{\vec{u} \in \Delta(\vec{a}):u_i=j} |\rho_{\vec{u}}|^p$. Again by By

737    Proposition B.7, with probability $1 - 1/\mathrm{poly}(n)$, we will have $|W_j^{i,k}|^p$ is a $O((c \log(n))^q)$-error

738    estimate of $\sum_{\vec{u} \in \Delta(\vec{a}):u_i=j} |\rho_{\vec{u}}|^p$ or at least one $k \in [\tau]$. Our goal will now be to find all $j \in [n_i]$

739    such that $\sum_{\vec{u} \in \Delta(\vec{a}):u_i=j} |\rho_{\vec{u}}|^p \geq \Theta((c \log(n))^q/r_3^8) \sum_{\vec{u} \in \Delta(\vec{a})} |\rho_{\vec{u}}|^p$. We call such a $j$ a *heavy hitter*.

740    Let $Q_i \subset [n_i]$ be the set of heavy hitters. To find all the heavy hitters, we use the count-sketch

741    for heavy hitters with the Dyadic trick of definition B.6. We construct this count-sketch of def B.6

742    $S^i \in \mathbb{R}^{k' \times n_i}$ where $k' = O(\log^2(n) r_3^{16})$. We then compute $S^i W^{i,k}$, for $k = 1, 2, \ldots, \tau$, and obtain

743    the set of heavy hitters $h \in H_{i,k} \subset [n_i]$ which satisfy $|W_j^{i,k}|^p \geq \Theta(1/r_3^8)\|W^{i,k}\|_p^p$. By the above

744    discussion, we know that for each $j \in Q_i$, we will have $|W_j^{i,k}|^p \geq \Theta(1/r_3^{16})\|W^{i,k}\|_p^p$ for at least

745    one $k \in [\tau]$ with high probability. Thus $H_i = \cup_{k=1}^{\tau} H_{i,k} \supseteq Q_i$.

746    We now will decide to either sample a heavy hitter $\xi \in H_i$, or a non-heavy hitter $\xi \in$

747    $[n_i] \setminus H_i$. By Proposition B.7, we can compute a $O((c \log(n))^{-q})$-error estimate $\beta_i =$

748    $\mathrm{median}_{j \in \tau} \left|\left((\bigotimes_{k=1}^{i-1} e_{a_k}^{\top}) \otimes (\bigotimes_{k=i}^{q} Z^{k,j})\rho\right)\right|^p$ of $\sum_{\vec{u} \in \Delta(\vec{a})} |\rho_{\vec{u}}|^p$, meaning:

$$O(c^{-q}) \sum_{\vec{u} \in \Delta(\vec{a})} |\rho_{\vec{u}}|^p \leq \beta_i \leq O((c \log n)^q) \sum_{\vec{u} \in \Delta(\vec{a})} |\rho_{\vec{u}}|^p.$$

749    Again, by Proposition B.7, we can compute a $O((c \log(n))^{-q})$-error estimate $\gamma_i \leftarrow$

750    $\mathrm{median}_{j \in [\tau]} \left((\bigotimes_{k=1}^{i-1} e_{a_k}^{\top}) \otimes Z_{[n_i] \setminus H_i}^{i,j} \otimes (\bigotimes_{k=i+1}^{q} Z^{k,j})\rho\right)$ of $\sum_{h \in [n_i] \setminus H_i} \sum_{\vec{u} \in \Delta(\vec{a}):u_i=j} |\rho_{\vec{u}}|^p$. It

751    follows that

$$O(c^{-2q}) \frac{\sum_{h \in [n_i] \setminus H_i} \sum_{\vec{u} \in \Delta(\vec{a}):u_i=j} |\rho_{\vec{u}}|^p}{\sum_{\vec{u} \in \Delta(\vec{a})} |\rho_{\vec{u}}|^p} \leq \frac{\gamma_i}{\beta_i} \leq O((c \log n)^{2q}) \frac{\sum_{h \in [n_i] \setminus H_i} \sum_{\vec{u} \in \Delta(\vec{a}):u_i=j} |\rho_{\vec{u}}|^p}{\sum_{\vec{u} \in \Delta(\vec{a})} |\rho_{\vec{u}}|^p}$$

752    In other words, $\gamma_i/\beta_i$ is a $O((c \log(n))^{2q})$-error approximation of the true probability that we should

753    sample a non-heavy item. Thus with probability $1 - \gamma_i/\beta_i$, we choose to sample a heavy item.

754    To sample a heavy item, for each $\xi \in H_i$, by Proposition B.7, we can compute an $O((c \log(n))^{-q})$-

755    error estimate $\mathrm{median}_{j \in \tau} \left|\left((\bigotimes_{k=1}^{i-1} e_{a_k}^{\top}) \otimes (e_{\xi}^{\top}) \otimes (\bigotimes_{k=i+1}^{q} Z^{k,j})\rho\right)\right|^p$ of $\sum_{\vec{u} \in \Delta(\vec{a}):u_i=\xi} |\rho_{\vec{u}}|^p$,

756    meaning

$$\left( O(c^{-q}) \sum_{\vec{u} \in \Delta(\vec{a}): u_i = \xi} |\rho_{\vec{u}}|^p \right) \le \text{median}_{j \in \tau} \left| \left( (\bigotimes_{k=1}^{i-1} e_{a_k}^\top) \otimes (e_\xi^\top) \otimes (\bigotimes_{k=i+1}^{q} Z^{k,j}) \rho \right) \right|^p$$

$$\le \left( O((c \log n)^q) \sum_{\vec{u} \in \Delta(\vec{a}): u_i = \xi} |\rho_{\vec{u}}|^p \right)$$

757  Thus we can choose to sample a heavy item $\xi \in H_i$ from the distribution given by

$$\Pr\left[\text{sample } a_i \leftarrow \xi\right] = \frac{\text{median}_{j \in \tau} \left| \left( (\bigotimes_{k=1}^{i-1} e_{a_k}^\top) \otimes (e_\xi^\top) \otimes (\bigotimes_{k=i+1}^{q} Z^{k,j}) \rho \right) \right|^p}{\sum_{\xi' \in H_i} \text{median}_{j \in \tau} \left| \left( (\bigotimes_{k=1}^{i-1} e_{a_k}^\top) \otimes (e_{\xi'}^\top) \otimes (\bigotimes_{k=i+1}^{q} Z^{k,j}) \rho \right) \right|^p}$$

758  Which gives a $O((c \log(n))^{2q})$-error approximation to the correct sampling probability for a heavy
759  item.

760  In the second case, with probability $\gamma_i/\beta_i$, we choose to not sample a heavy item. In this case, we
761  must now sample a item from $[n_i] \setminus H_i$. To do this, we partition $[n_i]$ randomly into $\Omega_1, \dots, \Omega_\eta$ for
762  $\eta = 1/r_3^2$. Now there are two cases. First suppose that we have

$$\frac{\sum_{j \in [n_i] \setminus H_i} \sum_{\vec{u} \in \Delta(\vec{a}): u_i = j} |\rho_{\vec{u}}|^p}{\sum_{\vec{u} \in \Delta(\vec{a})} |\rho_{\vec{u}}|^p} \le \Theta(1/r_3^3)$$

763  Now recall that $\gamma_i/\beta_i$ was a $O((c \log(n))^{2q})$-error estimate of the ratio on the left hand side
764  of the above equation, and $\gamma_i/\beta_i$ was the probability with which we choose to sample a
765  non-heavy hitter. Since we only repeat the sampling process $r_3$ times, the probability that
766  we ever sample a non-heavy item in this case is at most $\Theta(q(c \log(n))^{2q}/r_3^2) < \Theta(q/r_3)$,
767  taken over all possible repetitions of this sampling in the algorithm. Thus we can safely
768  ignore this case, and condition on the fact that we never sample a non-heavy item in this
769  case. Otherwise, $\sum_{j \in [n_i] \setminus H_i} \sum_{\vec{u} \in \Delta(\vec{a}): u_i = j} |\rho_{\vec{u}}|^p > \Theta(1/r_3^3) \sum_{\vec{u} \in \Delta(\vec{a})} |\rho_{\vec{u}}|^p$, and it follows that
770  $\sum_{\vec{u} \in \Delta(\vec{a}): u_i = j'} |\rho_{\vec{u}}|^p \le \Theta(1/r_3^5 \sum_{j \in [n_i] \setminus H_i}) \sum_{\vec{u} \in \Delta(\vec{a}): u_i = j} |\rho_{\vec{u}}|^p$ for all $j' \in [n_i] \setminus H_i$, since we
771  removed all $\Theta(1/r_3^3)$ heavy hitters from $[n_i]$ originally. Thus by Chernoff bounds, with high prob-
772  ability we have that $\sum_{j \in \Omega_i \setminus H_i} (\sum_{\vec{u} \in \Delta(\vec{a}): u_i = j} |\rho_{\vec{u}}|^p) = \Theta(1/\eta \sum_{j \in [n_i] \setminus H_i} \sum_{\vec{u} \in \Delta(\vec{a}): u_i = j} |\rho_{\vec{u}}|^p)$,
773  which we can union bound over all repetitions.

774  Given this, by choosing $t \sim [\eta]$ uniformly at random, and then choosing $j \in \Omega_t \setminus H_i$ with
775  probability proportional to its mass in $\Omega_t \setminus H_i$, we get a $\Theta(1)$ approximation of the true sam-
776  pling probability. Since we do not know its exact mass, we instead sample from the distribution
777  $\{\frac{\theta_j}{\sum_{j' \in \Omega_t \setminus H_i} \theta_{j'}}\}_{j \in \Omega_t \setminus H_i}$, where

$$\theta_j = \text{median}_{l \in [\tau]} \left( \left| (\bigotimes_{k=1}^{i-1} e_{a_k}^\top) \otimes (e_j^\top) \otimes (\bigotimes_{k=i+1}^{q} Z^{k,l}) \rho \right|^p \right)$$

778  Again by Proposition B.7, this gives a $O((c \log(n))^{2q})$-error approximation to the correct sam-
779  pling probability. Note that at each step of sampling a coorindate of $\vec{a}$ we obtained at most
780  $O((c \log(n))^{2q})$-error in the sampling probability. Thus, by oversampling by a $O((c \log(n))^{2q^2})$
781  factor, we can obtain the desired sampling probabilities. This completes the proof of correctness.
782  Note that to improve the failure probability to $1 - \delta$, we can simply scale $r_3$ by a factor of $1/\delta$.

783  **Proof of Runtime.** We now analyze the runtime. At every step $i = 1, 2, \dots, q$ of the sampling, we
784  compute $v_i^j \leftarrow S^i \left( (\bigotimes_{k=1}^{i-1} e_{a_k}^\top) \otimes (\mathbb{I}_{n_i}) \otimes (\bigotimes_{k=i+1}^{q} Z^{k,j}) \rho \right) \in \mathbb{R}^{n_i}$ for $j = 1, 2, \dots \Theta(\log(n))$.
785  This is equal to

$$S^i \left( (\bigotimes_{k=1}^{i-1} (A_k)_{a_k, *}) \otimes (A_i) \otimes (\bigotimes_{k=i+1}^{q} Z^{k,j} A_k) x' - (\bigotimes_{k=1}^{i-1} e_{a_k}^\top) \otimes (\mathbb{I}_{n_i}) \otimes (\bigotimes_{k=i+1}^{q} Z^{k,j}) b \right)$$

786   We first consider the term inside of the parenthesis (excluding $S^i$). Note that the term
787   $(\bigotimes_{k=i+1}^{q} Z^{k,j} A_k)$ was already pre-computed, and is a vector of length at most $d$, this this re-
788   quires a total of $\widetilde{O}(\sum_{i=1}^{q} \mathrm{nnz}(A_i) + d)$ time. Note that these same values are used for every
789   sample. Given this pre-computation, we can rearrage the first term to write $(\bigotimes_{k=1}^{i-1}(A_k)_{a_k,*}) \otimes$
790   $(A_i) X'(\bigotimes_{k=i+1}^{q} Z^{k,j} A_k)^\top$ where $X'$ is a matrix formed from $x'$ so that $x'$ is the vectorization of
791   $X'$ (this is done via reshaping Lemma B.2). The term $y = X'(\bigotimes_{k=i+1}^{q} Z^{k,j} A_k)^\top$ can now be
792   computed in $O(d)$ time, and then we reshape again to write this as $(\bigotimes_{k=1}^{i-1}(A_k)_{a_k,*}) Y A_i^\top$ where $Y$
793   again is a matrix formed from $y$. Observe that $\zeta = \mathrm{vec}(\bigotimes_{k=1}^{i-1}(A_k)_{a_k,*} Y) \in \mathbb{R}^{d_i}$ can be computed
794   in time $O(qd)$, since each entry is a dot product of a column $Y_{*,j} \in \mathbb{R}^{d_1 \cdot d_2 \cdots d_{i-1}}$ of $Y$ with the
795   $d_1 \cdot d_2 \cdots d_{i-1}$ dimensional vector $\bigotimes_{k=1}^{i-1}(A_k)_{a_k,*}$, which can be formed in $O(d_1 \cdot d_2 \cdots d_{i-1} q)$
796   time, and there are a total of $d_i$ columns of $Y$.

797   Given this, The first entire term $S^i(\bigotimes_{k=1}^{i-1}(A_k)_{a_k,*}) \otimes (A_i) \otimes (\bigotimes_{k=i+1}^{q} Z^{k,j} A_k) x'$ can be rewritten
798   as $S^i A_i \zeta$, where $\zeta = \zeta_{\vec{a}} \in \mathbb{R}^{d_i}$ can be computed in $O(dq)$ time for each sample $\vec{a}$. Thus if we
799   recompute the value $S_i A_i \in \mathbb{R}^{k \times n}$, where $k = \widetilde{O}(r_3^{16})$, which can be done in time $\widetilde{O}(\mathrm{nnz}\, A_i)$, then
800   every time we are sampling the $i$-th coordinate of some $\vec{a}$, computing the value of $S^i A_i \zeta_{\vec{a}}$ can be
801   done in time $O(k d_i^2) = r_3^{O(1)}$.

802   We now consider the second term. We do a similar trick, reshaping $b \in \mathbb{R}^n$ into $B \in$
803   $\mathbb{R}^{(n_1 \cdots n_i) \times (n_i \cdots n_q)}$ and writing this term as $((\bigotimes_{k=1}^{i-1} e_{a_k}^\top) \otimes (\mathbb{I}_{n_i})) B(\bigotimes_{k=i+1}^{q} Z^{k,j})^\top$ and computing
804   $b' = B(\bigotimes_{k=i+1}^{q} Z^{k,j})^\top \in \mathbb{R}^{(n_1 \cdots n_i)}$ in $\mathrm{nnz}(B) = \mathrm{nnz}(b)$ time. Let $B' \in \mathbb{R}^{(n_1 \cdots n_{i-1}) \times n_i}$ be such
805   that $\mathrm{vec}(B') = b'$, and we reshape again to obtain $(\bigotimes_{k=1}^{i-1} e_{a_k}^\top) B'(\mathbb{I}_{n_i}) = (\bigotimes_{k=1}^{i-1} e_{a_k}^\top) B'$ Now note
806   that so far, the value $B'$ did not depend on the sample $\vec{a}$ at all. Thus for each $i = 1, 2, \ldots, q$, $B'$
807   (which depends only on $i$) can be pre-computed in $\mathrm{nnz}(b)$ time. Given this, the value $(\bigotimes_{k=1}^{i-1} e_{a_k}^\top) B'$
808   is just a row $B'_{(a_1, \ldots, a_k), *}$ of $B'$ (or a column of $(B')^\top$). We first claim that $nnz(B') \leq \mathrm{nnz}(b) =$
809   $\mathrm{nnz}(B)$. To see this, note that each entry of $B'$ is a dot product $B_{j,*}(\bigotimes_{k=i+1}^{q} Z^{k,j})^\top$ for some
810   row $B_{j,*}$ of $B$, and moreover there is a bijection between these dot products and entries of $B'$.
811   Thus for every non-zero entry of $B'$, there must be a unique non-zero row (and thus non-zero en-
812   try) of $B$. This gives a bijection from the support of $B'$ to the support of $B$ (and thus $b$) which
813   completes the claim. Since $S^i(B'_{(a_1, \ldots, a_k), *})^\top$ can be computed in $\widetilde{O}(\mathrm{nnz}(B'_{(a_1, \ldots, a_k), *}))$ time, it
814   follows that $S^i(B'_{(a_1, \ldots, a_k), *})^\top$ can be computed for all rows $(B'_{(a_1, \ldots, a_k), *})$ of $B$ in $\widetilde{O}(\mathrm{nnz}(b))$
815   time. Given this precomputation, we note that $(\mathbb{I}_{n_i}) \otimes (\bigotimes_{k=i+1}^{q} Z^{k,j}) b$ is just $S^i(B'_{(a_1, \ldots, a_k), *})^\top$
816   for some $(a_1, \ldots, a_k)$, which has already been pre-computed, and thus requires no addition time per
817   sample. Thus, given a total of $\widetilde{O}(\sum_{i=1}^{q} \mathrm{nnz}(A_i) + q\,\mathrm{nnz}(b) + r_3^{O(1)})$ pre-processing time, for each
818   sample we can compute $v_i^j$ for all $i \in [q]$ and $j \in [\tau]$ in $\widetilde{O}(r_3^{O(1)})$ time, and thus $\widetilde{O}(r_3^{O(1)})$ time over
819   all $r_3$ samples.

820   Given this, the procedure to compute the heavy hitters $H_{i,j}$ takes $\widetilde{O}(r_3^{16})$ time by Definition B.6 for
821   each sample and $i \in [q], j \in [\tau]$. By a identical pre-computation and rearrangement argument as
822   above, each $\beta_i^j$ (and thus $\beta_i$) can be computed in $\widetilde{O}(r_3^{O(1)})$ time per sample after pre-computation.
823   Now note that $\gamma_i$ is simply equal to $\mathrm{median}\, j \in [\tau](\beta_i^j - (\bigotimes_{k=1}^{i-1} e_{a_k}^\top) \otimes (Z_{H_i}^{k,j}) \otimes (\bigotimes_{k=i+1}^{q} Z^{k,j})\rho)$.
824   Since $(Z_{H_i}^{k,j})$ is sparse, the above can similar be computed in $O(d|H_i|) = \widetilde{O}(r_3^{O(1)})$ time per sample
825   after pre-computation. To see this, note that the $b$ term of $(\bigotimes_{k=1}^{i-1} e_{a_k}^\top) \otimes (Z_{H_i}^{k,j}) \otimes (\bigotimes_{k=i+1}^{q} Z^{k,j})\rho$
826   can be written as $(\bigotimes_{k=1}^{i-1} e_{a_k}^\top) B'''(Z_{H_i}^{k,j})^\top$, where $B''' \in \mathbb{R}^{n_1 \cdots n_{i-1} \times n_i}$ is a matrix that has already
827   been pre-computed and does not depend on the given sample. Then this quantity is just the dot
828   product of a row of $B'''$ with $(Z_{H_i}^{k,j})^\top$, but since $(Z_{H_i}^{k,j})$ is $|H_i|$-sparse, so the claim for the $b$ term
829   follows. For the $(A_1 \otimes \cdots \otimes A_q)$ term, just as we demonstrated in the discussion of computing $v_i^j$,
830   note that this can be written as $(\bigotimes_{k=1}^{i-1}(A_k)_{a_k,*}) Y((A_i)_{H_i,*})^\top$ for some matrix $Y \in \mathbb{R}^{d_1 \cdots d_i \times d_{i-1}}$
831   that has already been precomputed. Since $(A_i)_{H_i,*}$ only has $O(|H_i|)$ non-zero rows, this whole
832   product can be computed in time $O(d|H_i|)$ as needed.

833 Similarly, we can compute the sampling probabilities

$$\Pr\left[\text{sample } a_i \leftarrow j\right] = \frac{\text{median}_{j\in\tau}\left|\left((\bigotimes_{k=1}^{i-1} e_{a_k}^\top) \otimes (e_\xi^\top) \otimes (\bigotimes_{k=i+1}^{q} Z^{k,j})\rho\right)\right|^p}{\sum_{\xi'\in H_i}\text{median}_{j\in\tau}\left|\left((\bigotimes_{k=1}^{i-1} e_{a_k}^\top) \otimes (e_{\xi'}^\top) \otimes (\bigotimes_{k=i+1}^{q} Z^{k,j})\rho\right)\right|^p}$$

834 for each every item $\zeta \in H_i$ in $\widetilde{O}(r_3^{O(1)})$ time after pre-computation, and note $|H_i| = \widetilde{O}(r_3^{O(1)})$ by
835 definition B.6. Thus the total time to sample a heavy hitter in a given coordinate $i \in [q]$ for each
836 sample $\widetilde{O}(r_3^{O(1)})$ per sample, for an overall time of $\widetilde{O}(qr_3^{O(1)})$ over all samples and $i \in [q]$.

837 Finally, we consider the runtime for sampling a non-heavy item. Note that $|\Omega_t| = O(n_i/\eta)$ with
838 high probability for all $t \in [\eta]$ by chernoff bounds. Computing each

$$\theta_j = \text{median}_{l\in[\tau]}\left(\left|(\bigotimes_{k=1}^{i-1} e_{a_k}^\top) \otimes (e_j^\top) \otimes (\bigotimes_{k=i+1}^{q} Z^{k,l})\rho\right|^p\right)$$

839 takes $O(qd)$ time after pre-computation, and so we spend a total of $O(qdn_i/\eta)$ time sampling an
840 item from $\Omega_t \setminus H_i$. Since we only ever sample a total of $r_3$ samples, and $\eta = \Theta(r_3^2)$, the total time for
841 sampling non-heavy hitters over the course of the algorithm in coordinate $i$ is $o(n_i) = o(\text{nnz}(A_i))$
842 as needed, which completes the proof of the runtime.

843 **Computing the Sampling Probabilities** $\alpha_i$  The above arguments demonstrate how to sample
844 efficiently from the desired distribution. We now must describe how the sampling probabilities $\alpha_i$
845 can be computed. First note, for each sample that is sampled in the above way, at every step we
846 compute exactly the probability with which we decide to sample a coordinate to that sample. Thus
847 we know exactly the probability that we choose a sample, and moreover we can compute each $q_i$ in
848 $O(d)$ time as in Lemma B.5. Thus we can compute the maximum of $q_i$ and this probability exactly.
849 For each item sampled as a result of the leverage score sampling probabilities $q_i$ as in Lemma B.5,
850 we can also compute the probability that this item was sampled in the above procedure, by using the
851 same sketching vectors $Z^{i,k}$ and count-sketches $S^i$. This completes the proof of the Lemma.

852 $\square$

## C  Missing Proofs from Section 4

854 In this section, we prove the correctness of our all-pairs regression algorithm 3. Our main theorem,
855 Theorem 4.1, relies crucially on the sample routine developed in Section C.1. We first prove the
856 theorem which utilizes this routine, and defer the description and proof of the routine to Section
857 C.1.

858 Recall first the high level description of our algorithm (given formally in Figure 3). We pick $S_1, S_2 \in$
859 $\mathbb{R}^{k\times n}$ and $S$ are sparse $p$-stable sketches. We then compute $M = (S_1 \otimes S_2)(F \otimes \mathbf{1} - \mathbf{1} \otimes F) =$
860 $S_1F \otimes S_2\mathbf{1} - S_1\mathbf{1} \otimes S_2F$, where $F = [A, b]$. We then take the $QR$ decomposition $M = QR$. Finally,
861 we sample rows of $(F \otimes \mathbf{1} - \mathbf{1} \otimes F)R^{-1}$ with probability proportional to their $\ell_p$ norms. This is
862 done by the sampling procedure described in Section C.1. Finally, we solve the regression problem
863 $\min_x \|\Pi(\bar{A}x - \bar{b})\|_p$, where $\Pi$ is the diagonal row-sampling matrix constructed by the sampling
864 procedure.

865 We begin by demonstrating that $S_1 \otimes S_2$ is a $\text{poly}(d)$ distortion embedding for the column span of
866 $[\bar{A}, \bar{b}]$.

867 **Lemma C.1.** *Let $S_1, S_2 \in \mathbb{R}^{k\times n}$ be sparse $p$-stable transforms, where $k = \text{poly}(d/(\epsilon\delta))$. Then for*
868 *all $x \in \mathbb{R}^{d+1}$, with probability $1 - \delta$ we have*

$$1/O(d^4 \log^4 d)\|[\bar{A}, \bar{b}]x\|_p \leq \|(S_1 \otimes S_2)[\bar{A}, \bar{b}]x\|_p \leq O(d^2 \log^2 d)\|[\bar{A}, \bar{b}]x\|_p$$

869 *Proof.* Let $F = [A, b]$. Then a basis for the columns of $[\bar{A}, \bar{b}]$ is given by $F \otimes \mathbf{1} - \mathbf{1} \otimes F$. We first
870 condition on both $S_1, S_2$ being a low-distortion embedding for the $d + 2$ dimensional column-span
871 of $[F, \mathbf{1}]$. Note that this holds with large constant probability by 2.3.

872 So for any $x \in \mathbb{R}^{d+1}$, we first show the upper bound

$$\|(S_{\mathbf 1} \otimes S_2)(F \otimes \mathbf{1} - \mathbf{1} \otimes F)x\|_p = \|(S_1 F \otimes S_2 \mathbf{1})x - (S_1 \mathbf{1} \otimes S_2 F)\|_p$$
$$= \|S_1 F x \mathbf{1}^\top S_2^\top - S_1 \mathbf{1} x^\top F^\top S_2^\top\|_p$$
$$= \|S_1 (F x \mathbf{1}^\top - \mathbf{1} x^\top F^\top) S_2^\top\|_p$$
$$\le O(d \log d)\|(F x \mathbf{1}^\top - \mathbf{1} x^\top F^\top) S_2^\top\|_p$$
$$\le O(d^2 \log^2 d)\|F x \mathbf{1}^\top - \mathbf{1} x^\top F^\top\|_p$$
$$= O(d^2 \log^2 d)\|(F \otimes \mathbf{1} - \mathbf{1} \otimes F)x\|_p$$

873   Where the first equality follows by properties of the Kronecker product [VL00], the second by
874   reshaping Lemma B.2. The first inequality follows from the fact that each column of $(F x \mathbf{1}^\top -$
875   $\mathbf{1} x^\top F^\top) S_2^\top$ is a vector in the column span of $[F, \mathbf{1}]$, and then using that $S_1$ is a low distortion
876   embedding. The second inequality follows from the fact that each row of $(F x \mathbf{1}^\top - \mathbf{1} x^\top F^\top)$ is
877   a vector in the *column span* of $[F, \mathbf{1}]$, and similarly using that $S_2$ is a low distortion embedding.
878   The final inequality follows from reshaping. Using a similar sequence of inequalities, we get the
879   matching lower bound as desired.        $\square$

880   We now prove our main theorem.

881   **Theorem 4.1**   *Given $A \in \mathbb{R}^{n \times d}$ and $b \in \mathbb{R}^n$, for $p \in [1, 2]$ there is an algorithm for the All-Pairs*
882   *Regression problem that outputs $\widehat{x} \in \mathbb{R}^d$ such that with probability $1 - \delta$ we have*

$$\|\bar{A}\widehat{x} - \bar{b}\|_p \le (1 + \epsilon) \min_{x \in \mathbb{R}^d} \|\bar{A}x - \bar{b}\|_p$$

883   *Where $\bar{A} = A \otimes \mathbf{1} - \mathbf{1} \otimes A \in \mathbb{R}^{n^2 \times d}$ and $\bar{b} = b \otimes \mathbf{1} - \mathbf{1} \otimes b \in \mathbb{R}^{n^2}$. For $p < 2$, the running time is*
884   $\widetilde{O}(nd + (d/(\epsilon\delta))^{O(1)})$, *and for $p = 2$ the running time is $O(\mathrm{nnz}(A) + (d/(\epsilon\delta))^{O(1)})$.*

885   *Proof.* We first consider the case of $p = 2$. Here, we can use the fact that the TENSORSKETCH
886   random matirx $S \in \mathbb{R}^{k \times n}$ is a subspace embedding for the column span of $[\bar{A}, \bar{b}]$ when $k = \Theta(d/\epsilon^2)$
887   [DSSW18], meaning that $\|S[\bar{A}, \bar{b}]\|_2 = (1 \pm \epsilon)\|[\bar{A}, \bar{b}]x\|_2$ for all $x \in \mathbb{R}^{d+1}$ with probability $9/10$.
888   Moreover, $S\bar{A}$ and $S\bar{b}$ can be computed in $O(\mathrm{nnz}(A) + \mathrm{nnz}(b)) = O(\mathrm{nnz}(A))$ by [DSSW18]
889   since they are the difference of Kronecker products. As a result, we can simply solve the regression
890   problem $\widehat{x} = \arg\min_x \|S\bar{A}x - S\bar{b}\|_2$ in $\mathrm{poly}(kd)$ time to obtain the desired $\widehat{x}$.

891   For $p < 2$, we use the algorithm in Figure 3, where the crucial leverage score sampling procedure
892   to obtain $\Pi$ in step 7 of Figure 3 is described in Lemma 4.2. Our high level approach follows the
893   general $\ell_p$ sub-space embedding approach of [DDH+09]. Namely, we first compute a low-distortion
894   embedding $(S_1 \otimes S_2)(F \otimes \mathbf{1} - \mathbf{1} \otimes F)$. By Lemma C.1, using sparse-p stable transformations
895   $S_1, S_2$, we obtain the desired $\mathrm{poly}(d)$ distortion embedding into $\mathbb{R}^{k^2}$, where $k = \mathrm{poly}(d/\epsilon)$. Note
896   that computing $(S_1 \otimes S_2)(F \otimes \mathbf{1} - \mathbf{1} \otimes F)$ can be done in $O(\mathrm{nnz}(A) + \mathrm{nnz}(b) + n)$ time using
897   the fact that $(S_1 \otimes S_2)(F \otimes \mathbf{1}) = S_1 F \otimes S_2 \mathbf{1}$. As shown in [DDH+09], it follows that $M =$
898   $(F \otimes \mathbf{1} - \mathbf{1} \otimes F)R^{-1}$ is an $\ell_p$ well-conditioned basis for the column span of $(F \otimes \mathbf{1} - \mathbf{1} \otimes F)$ (see
899   definition 2.5). Then by Theorem 5 of [DDH+09], if we let $\widehat{\Pi}$ be the diagonal row sampling matrix
900   such that $\widehat{\Pi}_{i,i} = 1/q_i^{1/p}$ for each $i$ with probability $q_i \ge \min\{1, r\|M_{i,*}\|_p^p/\|M\|_p^p\}$ (and $\widehat{\Pi}_{i,i} = 0$
901   otherwise) for $r = \mathrm{poly}(d \log(1/\delta)/\epsilon)$, then with probability $1 - \delta$ we have $\|\widehat{\Pi}(F \otimes \mathbf{1} - \mathbf{1} \otimes F)x\|_p =$
902   $(1 \pm \epsilon)\|(F \otimes \mathbf{1} - \mathbf{1} \otimes F)x\|_p$ for all $x \in \mathbb{R}^{d+1}$. First assume that we had such a matrix.

903   Since $(\bar{A}x - \bar{b})$ is in the column span of $(F \otimes \mathbf{1} - \mathbf{1} \otimes F)$ for any $x \in \mathbb{R}^{d+1}$, it follows that
904   $\|\widehat{\Pi}(\bar{A}x - \bar{b})\|_p = (1 \pm \epsilon)\|(\bar{A}x - \bar{b})\|_p$ for all $x \in \mathbb{R}^d$, which completes the proof of correctness.
905   By Lemma 4.2, we can obtain a row sampling matrix $\Pi$ in time $\widetilde{O}(nd + \mathrm{poly}(d/\epsilon))$, except that
906   the entries of $\Pi$ are instead equal to either 0 or $1/\widetilde{q}_i^{1/p}$ where $\widetilde{q}_i = (1 \pm \epsilon^2)q_i$. Now let $\widehat{\Pi}$ be the
907   idealized row sampling matrices from above, with entries either 0 or $1/q_i^{1/p}$ as needed for Theorem
908   5 of [DDH+09]. Note that for any matrix $Z$ each row of $\widehat{\Pi}Zx$ is equal to $\Pi Zx$ times some constant
909   $1 - \epsilon^2 < c < 1 + \epsilon^2$. It follows that $\|\Pi(\bar{A}x - \bar{b})\|_p = (1 \pm \epsilon^2)\|\widehat{\Pi}(\bar{A}x - \bar{b})\|_p$ for all $x \in \mathbb{R}^d$, and thus

910 the objective function is changed by at most a $(1 \pm \epsilon^2)$ term, which is simply handled by a constant
911 factor rescaling of $\epsilon$.

912 Finally, we can solve the sketched regression problem $\|\Pi(\bar{A}x - \bar{b})\|_p$ which has $\text{poly}(d/\epsilon)$ con-
913 straints and $d$ variables in time $\text{poly}(d/\epsilon)$ using linear programming for $p = 1$ (see [CLS19] for the
914 state of the art linear program sovler), or more generally interior point methods for convex program-
915 ming for $p > 1$ (see [BCLL18] for the state of the art $\ell_p$ solver. Finally, the failure probability bound
916 holds by union bounding over all the aforementioned results, and noting that the lowest probability
917 event was the even that $S_1 \otimes S_2$ was a low distortion embedding via Lemma C.1. This completes
918 the proof of the theorem.

919 $\qquad\qquad\qquad\qquad\qquad\qquad\qquad\qquad\qquad\qquad\qquad\qquad\qquad\qquad\qquad\qquad\qquad\qquad\qquad\qquad\qquad\qquad\qquad\qquad\qquad\qquad\qquad\qquad\qquad$ $\square$

## C.1  Proof of Fast Sampling Lemma 4.2

921 We now provide a full proof of the main technical lemma of Section 4. The sampling algorithm is
922 given formally in Algorithm 5. The following proof of Lemma 4.2 analyzes each step in the process,
923 demonstrating both correctness and the desired runtime bounds.

924 **Lemma 4.2***Given $R \in \mathbb{R}^{(d+1) \times (d+1)}$ and $F = [A, b] \in \mathbb{R}^{n \times (d+1)}$, there is an algorithm that, with*
925 *probability $1 - \delta$ for any $\delta > n^{-c}$ for any constant $c$, produces a diagonal matrix $\Pi \in \mathbb{R}^{n^2 \times n^2}$ such*
926 *that $\Pi_{i,i} = 1/\widetilde{q}_i^{1/p}$ with probability $q_i \geq \min\{1, r\|M_{i,*}\|_p^p/\|M\|_p^p\}$ and $\Pi_{i,i} = 0$ otherwise, where*
927 *$r = \text{poly}(d/\epsilon)$ and $M = (F \otimes \mathbf{1} - \mathbf{1} \otimes F)R^{-1}$, and $\widetilde{q}_i = (1 \pm \epsilon^2)q_i$ for all $i \in [n^2]$. The total time*
928 *required is $\widetilde{O}(\text{nnz } A + \text{poly}(d/\epsilon))$.*

929 *Proof.* Our proof proceeds in several steps. We analyze the runtime concurrently with out analysis
930 of correctness.

931 **Reducing the number of Columns of $R^{-1}$**  We begin by generating a matrix $G \in \mathbb{R}^{(d+1) \times \xi}$ of
932 i.i.d. $\mathcal{N}(0, 1/\sqrt{\xi})$ Gaussian random variables. We then compute $Y \leftarrow R^{-1}G$ in $\widetilde{O}(d^2)$ time. We
933 first claim that it suffices to instead $\ell_p$ sample rows of $C = (F \otimes \mathbf{1} - \mathbf{1} \otimes F)Y = MG$. Note that
934 each entry $|C_{i,j}|^p$ is distributed as $g^p\|M_{i,*}\|_2^p$ where $G$ $\mathcal{N}(0, 1/\sqrt{\xi})$ Gaussian, which holds by the
935 2-stability of Gaussian random variables. Note that $\mathbb{E}[|g|^p] = \Theta(1/\xi)$, so $\mathbb{E}[\|C_{i,*}\|_p^p] = \|M_{i,*}\|_2^p$,
936 and by sub-exponential concentration (see Chapter 2 of [Wai19]), we have that $\|C_{i,*}\|_p^p = (1 \pm$
937 $1/10)\|M_{i,*}\|_2^p$ with probability $1 - 1/\text{poly}(n)$, and we can union bound over this holding for all
938 $i \in [n^2]$. By relationships between the $p$ norms, we have $\|M_{i,*}\|_p^p/d < \|M_{i,*}\|_2^p < \|M_{i,*}\|_p^p$, thus
939 this changes the overall sampling probabilities by a factor between $\Theta(1/d^2)$ and $\Theta(d^2)$. Thus, we
940 can safely oversample by this factor (absorbing it into the value of $r$) to compensate for this change
941 in sampling probabilities.

942 **Sampling a row from $C$.**    To sample a row from $C$, the approach will be to sample an entry
943 $C_{i,j}$ of $C$ with probability proportional to $\|C_{i,j}\|_p^p/\|C\|_p^p$. For every $(i, j)$ sampled, we sample
944 the entire $i$-th row of $j$, so that the $j$-th row is indeed sampled with probability proportional to its
945 norm. Thus, it suffices to sample entries of $C$ such that each $C_{i,j}$ is chosen with probability at
946 least $\min\{1, r\|C_{i,j}\|_p^p/\|C\|_p^p\}$. First note that the $i$-th column of $C = (F \otimes \mathbf{1} - \mathbf{1} \otimes F)Y$ can be
947 rearranged into a $n \times n$ matrix via Lemma B.2, given by $(FY_{*,i}\mathbf{1}^\top - \mathbf{1}Y_{*,i}^\top F^\top)$. To $\ell_p$ sample a
948 coordinate from $C$, it suffices to first $\ell_p$ sample a column of one of the above matrices, and then $\ell_p$
949 sample an entry from that column.

950 To do this, we first compute $FY \in \mathbb{R}^{n \times \xi}$, which can be done in time $\widetilde{O}(\text{nnz } A)$ because $Y$ only has
951 $\xi = \Theta(\log(n))$ columns. We then compute $Z(FY_{*,i}\mathbf{1}^\top - \mathbf{1}Y_{*,i}^\top F^\top) \in \mathbb{R}^{1 \times n}$ for all $i \in [d]$, where
952 $Z \in \mathbb{R}^{1 \times n}$ is a fixed vector of i.i.d. $p$-stable random variables. Once $FY$ has been computed, for
953 each $i \in [\xi]$ it takes $O(n)$ time to compute this $n$-dimensional vector, thus the total time required
954 to compute all $\xi$ vectors is $\widetilde{O}(n)$. We repeat this process $t = O(\log(n))$ times with different $p$-
955 stable vectors $Z^1, \ldots, Z^\top$, and take the median of each coordinate of $Z^j(FY_{*,i}\mathbf{1}^\top - \mathbf{1}Y_{*,i}^\top F^\top) \in$
956 $\mathbb{R}^n$, $j \in [t]$, divided by the median of the $p$-stable distribution (which can be approximated to
957 $(1 \pm \epsilon)$ error in $\text{poly}(1/\epsilon)$ time, see Appendix A.2 of [KNW10] for details of this). This is done
958 in Step 7 of Algorithm 5. It is standard this this gives a $(1 \pm 1/10)$ approximation the the norm

**Algorithm 5** Algorithm to $\ell_p$ sample $\Theta(r)$ rows of $M = (F \otimes \mathbf{1} - \mathbf{1} \otimes F)R^{-1}$

1: **procedure** $\ell_p$ SAMPLE($F = [A, b] \in \mathbb{R}^{n \times d}, R^{-1} \in \mathbb{R}^{d+1 \times d+1}, r$)
2:     Generate a matrix $G \in \mathbb{R}^{d+1 \times \xi}$ of i.i.d. $\mathcal{N}(0, 1/\sqrt{\xi})$ Gaussian random variables, with $\xi = \Theta(\log(n))$
3:     $Y \leftarrow R^{-1}G \in \mathbb{R}^{d+1 \times \xi}$
4:     $C \leftarrow (F \otimes \mathbf{1} - \mathbf{1} \otimes F)Y$
5:     Reshape i-th column $C_{*,i}$ into $(FY_{*,i}\mathbf{1}^\top - \mathbf{1}(Y_{*,i})^\top F^\top) \in \mathbb{R}^{n \times n}$
6:     Generate $Z \in \mathbb{R}^{t \times n}$ i.i.d. $p$-stable for $t = \Theta(\log(n))$        $\triangleright$ Definition 2.1
7:     For all $(i, l) \in [\xi] \times [n]$, set

$$\sigma_{i,l} \leftarrow \underset{\tau \in [t]}{\mathrm{median}} \left( \frac{\left| (Z(FY_{*,i}\mathbf{1}^\top - \mathbf{1}(Y_{*,i})^\top F^\top)_{\tau,l} \right|^p}{(\mathrm{median}(\mathcal{D}_p))^p} \right)$$

                                             $\triangleright$ Indyk Estimator [Ind06]

8:     Set $W^{(i,l)} \leftarrow (FY_{*,i}\mathbf{1}^\top - \mathbf{1}Y_{*,i}^\top F^\top)_{*,l} = FY_{*,i} - \mathbf{1}(FY)_{l,i} \in \mathbb{R}^n$
9:     **for** $j = 1, \ldots, \Theta(r)$ **do**
10:         Sample $(i, l)$ from distribution $\sigma_{i,l} / \left( \sum_{i',l'} \sigma_{i',l'} \right)$.
11:     **end for**
12:     $T \leftarrow$ multi-set of samples $(i, l)$
13:     Generate $S_0 \in \mathbb{R}^{k \times n}$ $S \in \mathbb{R}^{k' \times n}$ count-sketches for heavy hitters with $k = r^{O(1)}, k' = k^{O(1)}$.                                                           $\triangleright$ Definition B.6
14:     Generate $u_1, \ldots, u_n$ i.i.d. exponential variables.
15:     $D \leftarrow \mathtt{Diag}(1/u_1^{1/p}, \ldots, 1/u_n^{1/p}) \in \mathbb{R}^{n \times n}$.
16:     **for** each sample $(i, l) \in T$ **do**
17:         Compute $S_0 W^{(i,l)}$ and obtain set of heavy hitters $Q_0^{(i,l)} \subset [n]$
18:         Compute $W_j^{(i,l)}$ exactly for all $j \in Q_0^{(i,l)}$, to obtain true heavy hitters $H^{(i,l)}$.
19:         Compute

$$\alpha_{i,l} \leftarrow \underset{\tau \in [t]}{\mathrm{median}} \left( \frac{\left| Z_{\tau,*}W^{(i,l)} - \sum_{\zeta \in H^{(i,l)}} Z_{\tau,\zeta}W_\zeta^{(i,l)} \right|^p}{(\mathrm{median}(\mathcal{D}_p))^p} \right)$$

20:         **if** With prob $1 - \alpha_{(i,l)}/\sigma_{(i,l)}$, sample a heavy item $j^* \leftarrow j$ **then**
21:             Sample a heavy item $j^* \leftarrow j$ from the distribution $|W_j^{(i,l)}|^p / \sum_{j \in H_{(i,l)}} |W_j^{(i,l)}|^p$.
22:             **return** The row $((l-1)n + j^*)$       $\triangleright$ Note that $C_{(l-1)n+j^*,*}$ contains $W_{j^*}^{(i,l)}$
23:         **else**
24:             Randomly partition $[n]$ into $\Omega_1, \Omega_2, \ldots, \Omega_\eta$ with $\eta = \Theta(r^4/\epsilon^4)$.
25:             Sample $t \sim [\eta]$ uniformly at random.
26:             Compute $S(DW^{(i,l)})_{\Omega_t \setminus H^{(i,l)}}$, and set $Q^{(i,l)} \subset \Omega_t \setminus H^{(i,l)}$ of heavy hitters.
27:             $j^* \leftarrow \arg\max_{j \in Q^{(i,l)}} (DW^{(i,l)})_j$
28:             **return** The row $((l-1)n + j^*)$       $\triangleright$ Note that $C_{(l-1)n+j^*,*}$ contains $W_{j^*}^{(i,l)}$
29:         **end if**
30:     **end for**
31: **end procedure**

$\|(FY_{*,i}\mathbf{1}^\top - \mathbf{1}Y_{*,i}^\top F^\top)_{*,l}\|_p$ for each $i \in [d], l \in [n]$ with probability $1 - 1/\mathrm{poly}(n)$ (See the Indyk median estimator [Ind06]).

Now let $\sigma_{i,l}$ be our estimate of the norm $\|(FY_{*,i}\mathbf{1}^\top - \mathbf{1}Y_{*,i}^\top F^\top)_{*,l}\|_p$, for all $i \in [\xi]$ and $l \in [n]$. We now sample a columns $(i, l) \in [\xi] \times [n]$, where each $(i, l)$ is chosen with probability $\sigma_{i,l}/(\sum_{i',l'} \sigma_{i',l'})$. We repeat this process $\Theta(r)$ times, to obtain a *multi-set* $T \subset [\xi] \times [n]$ of sampled columns $(i, l)$. We stress that $T$ is a multi-set, because the same column $(i, l)$ may have been chosen

965 for multiple samples, and each time it is chosen we must independently sample one of the entries of
966 that column. For any $(i, l) \in T$, we define $W^{(i,l)} = (FY_{*,i}\mathbf{1}^\top - \mathbf{1}Y_{*,i}^\top F^\top)_{*,l} = (FY_{*,i} - \mathbf{1}(FY)_{l,i})$.

967 $\ell_p$ **Sampling an entry from $W^{(i,l)}$.** Now fix any $(i, l) \in T$. We show how to $\ell_p$ sample an entry
968 from the vector $W^{(i,l)} \in \mathbb{R}^n$. In other words, for a given $j \in [n]$, we want to sample $W_j^{(i,l)} \in [n]$
969 with probability at least $r|W_j^{(i,l)}|^p/\|W^{(i,l)}\|_p^p$. We do this in two steps. First, let $S_0 \in \mathbb{R}^{k \times n}$
970 be the count-sketch for heavy hitters of definition B.6, where $k = \text{poly}(r)$. Note that we can
971 compute $S_0 FY$ and $S_0\mathbf{1}$ in time $\widetilde{O}(n)$, since $FY \in \mathbb{R}^{n \times \xi}$. Once this is done, for each $(i, l) \in$
972 $T$ we can compute $S_0 W^{(i,l)}$ in $O(k)$ time by computing $(S_0\mathbf{1}(FY)_{l,i})$ (note that $FY$ and $S_0\mathbf{1}$
973 are already computed), and subtracting it off from the $i$-th column of $S_0 FY$, so the total time is
974 $\widetilde{O}(n + \text{poly}(d/\epsilon))$ to compute $S_0 W^{(i,l)}$ for all $(i, l) \in |T|$. Now we can obtain the set $Q_0^{(i,l)} \subset [n]$
975 containing all the $\widetilde{\Omega}(1/\sqrt{k})$ heavy hitters in $W^{(i,l)}$ with high probability. We can then explicitly
976 compute the value of $W_j^{(i,l)}$ for all $j \in Q_0^{(i,l)}$, and exactly compute the set

$$H^{(i,l)} = \left\{ j \in [n] \ \Big| \ |W_j^{(i,l)}|^p > \beta/r^{16}\|W^{(i,l)}\|_p^p \right\},$$

977 all in $\widetilde{O}(k)$ time via definition B.6, where $\beta > 0$ is a sufficiently small constant (here we use the
978 fact that $|x|_p \geq |x|_2$ for $p \leq 2$). Note that we use the same sketch $S_0$ to compute all sets $Q_0^{(i,l)}$, and
979 union bound the event that we get the heavy hitters over all $\text{poly}(d/\epsilon)$ trails.

980 We are now ready to show how we sample an index from $W^{(i,l)}$. First, we estimate the total $\ell_p$
981 norm of the items in $[n_i] \setminus H^{(i,l)}$ (again with the Indyk median estimator), and call this $\alpha_{(i,l)}$ as in
982 Algorithm 5, which can be computed in $O(|H^{(i,l)}|)$ additional time (by subtracting off the $|H^{(i,l)}|$
983 coordinates $ZW_\zeta^{(i,l)}$ for all heavy hitters $\zeta \in H^{(i,l)}$ from our estimate $\sigma_{(i,l)}$), and with probability
984 $\alpha_{(i,l)}/\sigma_{(i,l)}$, we choose to sample one of the items of $H^{(i,l)}$, which we can then sample from the
985 distribution $|W_j^{(i,l)}|^p/(\sum_{j \in H^{(i,l)}} |W_j^{(i,l)}|^p)$. Since all the $\sigma_{(i,l)}, \alpha_{(i,l)}$'s were constant factor approx-
986 imations, it follows that we sampled such an item with probability $\Omega(r|W_{j'}^{(i,l)}|^p/\|C\|_p^p)$ as needed.
987 Otherwise, we must sample an entry from $[n] \setminus H^{(i,l)}$. To do this, we first randomly partition $[n]$
988 into $\eta = \Theta(r^4/\epsilon^4)$ subsets $\Omega_1, \Omega_2, \ldots, \Omega_\eta$.

989 We now make the same argument made in the proof of Lemma B.9, considering two cases. In the
990 first case, the $\ell_p$ mass of $[n] \setminus H^{(i,l)}$ drops by a $1/r^2$ factor after removing the heavy hitters. In
991 this case, $\alpha_{(i,l)}/\sigma_{(i,l)} = O(1/r^2)$, thus we will never *not* sample a heavy hitter with probability
992 $1 - O(1/r)$, which we can safely ignore. Otherwise, the $\ell_p$ drops by less than a $1/r^2$ factor, and it
993 follows that all remaining items must be at most a $\beta/r^{14}$ heavy hitter over the remaining coordinates
994 $[n] \setminus H^{(i,l)}$ (since if they were any larger, they would be $\beta/r^{16}$ heavy hitters in $[n]$, and would have
995 been removed in $H^{(i,l)}$). Thus we can assume we are in the second case. So by Chernoff bounds, we
996 have $\sum_{j \in \Omega_t} |W_j^{(i,l)}|_p = \Theta(\frac{1}{\eta} \sum_{j \in [n] \setminus H^{(i,l)}} |W_j^{(i,l)}|_p)$ with probability greater than $1 - \exp(-\Omega(r))$.
997 We can then union bound over this event occurring for all $t \in [\eta]$ and all $(i, l) \in T$. Given this, if
998 we uniformly sample a $t \sim [\eta]$, and then $\ell_p$ sample a coordinate $j \in \Omega_t$, we will have sampled this
999 coordinate with the correct probability up to a constant factor. We now sample such a $t$ uniformly
1000 from $\eta$.

1001 To do this, we generate a diagonal matrix $D \in \mathbb{R}^{n \times n}$, where $D_{i,i} = 1/u_i^{1/p}$, where $u_1, \ldots, u_n$
1002 are i.i.d. exponential random variables. For any set $\Gamma \subset [n]$, let $D_\Gamma$ be $D$ with all diagonal entries
1003 $(j, j)$ such that $j \notin \Gamma$ set equal to 0. Now let $S \in \mathbb{R}^{k' \times n}$ be a second instance of count-sketch for
1004 heavy hitters of definition B.6, where we set $k' = \text{poly}(k)$ from above. It is known that returning
1005 $j^* = \arg\max_{j \in \Omega_t \setminus H^{(i,l)}} |(DW^{(i,l)})_j|$ is a perfect $\ell_p$ sample from $\Omega_t \setminus H^{(i,l)}$ [JW18]. Namely,
1006 $\Pr[j^* = j] = |W_j^{(i,l)}|^p/\|W_{\Omega_t \setminus H^{(i,\ell)}}\|_p^p$ for any $j \in \Omega_t \setminus H^{(i,\ell)}$. Thus it will suffice to find this
1007 $j^*$. To find $j^*$, we compute $S(DW^{(i,l)})_{\Omega_t \setminus H^{(i,\ell)}}$. Note that since $FY$ has already been computed, to
1008 do this we need only compute $SD_{\Omega_t \setminus H^{(i,\ell)}}FY_{*,i}$ and $SD_{\Omega_t \setminus H^{(i,\ell)}}\mathbf{1}(FY)_{\ell,i}$, which takes total time
1009 $\widetilde{O}(|\Omega_t \setminus H^{(i,\ell)}|) = \widetilde{O}(n/\eta)$. We then obtain a set $Q^{(i,l)} \subset \Omega_t \setminus H^{(i,\ell)}$ which contains all $j$ with
1010 $|(DW^{(i,l)})_j| \geq \widetilde{\Omega}(1/\sqrt{k'})\|(DW)_{\Omega_t \setminus H^{(i,\ell)}}\|_2$.

As noted in [JW18], the value $\max_{j \in \Omega_t \setminus H^{(i,l)}} |(DW^{(i,l)})_j|$ is distributed identically to $\|W_{\Omega_t \setminus H^{(i,\ell)}}\|_p / u^{1/p}$ where $u$ is again an exponential random variable. Since exponential random variables have tails that decay like $e^{-\Omega(x)}$, it follows that with probability $1 - \exp(-\Omega(r))$ that we have $\max_{j \in \Omega_t \setminus H^{(i,l)}} |(DW^{(i,l)})_j| = \Omega(\|W_{\Omega_t \setminus H^{(i,\ell)}}\|_p / r)$, and we can then union bound over the event that this occurs for all $(i,l) \in T$ and $\Omega_t$. Given this it follows that $(DW^{(i,l)})_{j^*} = \Omega(\|W_{\Omega_t \setminus H^{(i,\ell)}}\|_p / r)$. Next, for any constant $c \geq 2$, by Proposition 1 of [JW18], we have $\|((DW)_{\Omega_t \setminus H^{(i,\ell)}})_{\mathtt{tail}(c \log(n))}\|_2 = \widetilde{O}(\|W^{(i,l)}_{\Omega_t \setminus H^{(i,\ell)}}\|_p)$ with probability $1 - n^{-c}$, where for a vector $x$, $x_{\overline{([t])}}$ is $x$ but with the top $t$ largest (in absolute value) entries set equal to 0. Since there are at most $c \log(n)$ coordinates in $(DW)_{\Omega_t \setminus H^{(i,\ell)}}$ not counted in $((DW)_{\Omega_t \setminus H^{(i,\ell)}})_{\mathtt{tail}(c \log(n))}$, and since $(DW)_{j^*}$ is the largest coordinate in all of $(DW)_{\Omega_t \setminus H^{(i,\ell)}}$, by putting together all of the above it follows that $(DW)_{j^*}$ is a $\widetilde{\Omega}(1/r)$-heavy hitter in $(DW)_{\Omega_t \setminus H^{(i,\ell)}}$. Namely, that $|(DW)_{j^*}| \geq \widetilde{\Omega}(\|(DW)_{\Omega_t \setminus H^{(i,\ell)}}\|_2 / r)$. Thus, we conclude that $j^* \in Q^{(i,l)}$.

Given that $j^* \in Q^{(i,l)}$, we can then compute the value $(DW^{(i,l)})_j = D_{j,j}(FY_{j,i} - FY_{l,i})$ in $O(1)$ time to find the maximum coordinate $j^*$. Since $|Q^{(i,l)}| = O(k') = O(\text{poly}(d/\epsilon))$, it follows that the total time required to do this is $\widetilde{O}(n/\eta + \text{poly}(d/\epsilon))$. Since we repeat this process for each $(i,l) \in T$, and $|T| = \Theta(r)$ whereas $\eta = \Theta(r^4)$, it follows that the total runtime for this step is $\widetilde{O}(n/r^3 + \text{poly}(d/\epsilon))$. By [JW18], the result is a perfect $\ell_p$ sample from $(DW)_{\Omega_t \setminus H^{(i,\ell)}}$, which is the desired result. To complete the proof, we note that the only complication that remains is that we utilize the same scaling matrix $D$ to compute the sampled used in each of the columns $W^{(i,l)}$ for each $(i,l) \in T$. However, note that for $t \neq t'$, we have that $D_{\Omega_t}$ and $D_{\Omega_t}$ are independent random variables. Thus it suffices to condition on the fact that the $t \in [\eta]$ that is sampled for each of the $|T|$ repetitions of sampling a $\Omega_t$ are distinct. But this occurs with probability at least $1/r$, since $|T| = \Theta(r)$ and $\eta = \Theta(r^4)$. Conditioned on this, all $|T|$ samples are independent, and each sample is an entry $C_{i,j}$ of $C$ such that the probability that a given $(i,j)$ is chosen is $|C_{i,j}|^p / \|C\|_p^p$. Repeating this sampling $\Theta(r)$ times, we get that each $C_{i,j}$ is sampled with probability at least $\min\{1, r|C_{i,j}|^p / \|C\|_p^p\}$, which completes the proof of correctness. Note that the dominant runtime of the entire procedure was $\widetilde{O}(\text{nnz}(A) + \text{poly}(d/\epsilon))$ as stated, and the probability of success was $1 - \exp(-r) + 1/\text{poly}(n)$, which we can be amplified to any $1 - \delta$ for $\delta > 1/n^c$ for some constant $c$ by increasing the value of $r$ by $\log(1/\delta)$ and the number of columns of the sketch $G$ to $\log(1/\delta)$, which does not effect the $\widetilde{O}(\text{nnz}(A) + \text{poly}(d/\epsilon))$ runtime.

**Computing approximations $\widetilde{q}_i$ for $q_i$.** It remains now how to compute the approximate sampling probabilities $\widetilde{q}_i$ for $\Theta(r)$ rows of $C$ that were sampled. Note that to sample an entry, in $C$, we first sampled the $n \times 1$ submatrix $W^{(i,l)}$ of $C$ which contained it, where the probability that we sample this submatrix is known to us. Next, if the entry of $C$ was a heavy hitter in $W^{(i,l)}$, we exactly compute the probability that we sample this entry, and sample it with this probability. If the entry $j$ of $W^{(i,l)}$ is not a heavy hitter, we first sample an $\Omega_t$ uniformly with probability exactly $1/\eta$. The last step is sampling a coordinate from $W^{(i,l)}_{\Omega_t \setminus H^{(i,l)}}$ via exponential scaling. However, we do not know the exact probability of this sampling, since this will be equal to $|W^{(i,l)}_j|^p / \|W^{(i,l)}_{\Omega_t \setminus H^{(i,l)}}\|_p^p$, and we do not know $\|W^{(i,l)}_{\Omega_t \setminus H^{(i,l)}}\|_p^p$ exactly. Instead, we compute it approximately to error $(1 \pm \epsilon^2)$ as follows. For each $(i,l) \in T$ and $\alpha = 1, 2, \ldots, \Theta(\log(n)/\epsilon^4)$, we compute $Z^{(\alpha)} W^{(i,l)}_{\Omega_t \setminus H^{(i,l)}}$, where $Z \in \mathbb{R}^{1 \times |\Omega_t \setminus H^{(i,l)}|}$ is a vector of $p$-stable random variables. Again, we use the Indyk median estimator [Ind06], taking the median of these $\Theta(\log(n)/\epsilon^4)$ repetitions, to obtain an estimate of $\|W^{(i,l)}_{\Omega_t \setminus H^{(i,l)}}\|_p^p$ with high probability to $(1 \pm \epsilon^2)$ relative error. Each repetition requires $O(|\Omega_t \setminus H^{(i,l)}|)$ additional time, and since $|\Omega_t \setminus H^{(i,l)}||T| = o(\epsilon^4 n/r^3)$, it follows that the total computational time is at most an additive $o(n)$, thus computing the $\widetilde{q}_i$'s to error $(1 \pm \epsilon^2)$ does not effect the overall runtime. $\square$

## D    Missing Proofs from Section 5

We first give the proof of the main theorem. The proof relies on Lemma D.4, which the rest of this section will be devoted to proving.

**Theorem 5.1**    *For any constant $q \geq 2$, there is an algorithm which runs in time $O(\sum_{i=1}^{q} \mathrm{nnz}(A_i) + d\,\mathrm{poly}(k/\epsilon))$ and outputs a rank $k$-matrix $B$ in factored form such that $\|B - A\|_F \leq (1 + \epsilon)\,\mathrm{OPT}_k$ with probability $9/10$.*

*Proof.* By Lemma D.4, we have $(1 - \epsilon)\|A - AP\|_F^2 \leq \|M - MP\|_F^2 + c \leq (1 + \epsilon)\|A - AP\|_F^2$ for all rank $k$ projection matrices $P$. In particular, we have

$$\min_{P}(1 + \epsilon)\|A - AP\|_F^2 + c = (1 + \epsilon)\,\mathrm{OPT}_k^2$$

where the minimum is taken over all rank $k$ projection matrices. The minimizer $P$ on the LHS is given by the projection onto the top $k$ singular space of $M$. Namely, $MP = MU^\top U$ where $U$ is the top $k$ singular row vectors of $M$. Thus $\|M - MU^\top U\|_F^2 + c \leq (1 + \epsilon)\,\mathrm{OPT}_k^2$. Moreover, we have $\|A - AU^\top U\|_F^2 \leq (1 + 2\epsilon)(\|M - MU^\top U\|_F^2 + c) \leq (1 + 4\epsilon)\,\mathrm{OPT}_k^2$. Thus $\|A - AU^\top U\|_F \leq (1 + O(\epsilon))\,\mathrm{OPT}_k$ as needed.

For runtime, note that we first must compute $M = (\otimes_{i=1}^{q} S_i)(A_1 \otimes A_2) = S_1 A_1 \otimes \cdots \otimes S_q A_q$. Now $S_i A_i$ can be computed in $O(\mathrm{nnz}(A_i))$ time for each $i$ [CW13]. One all $S_i A_i$ are computed, their Kronecker product can be computed in time $O(q k_1 k_2 \cdots k_q d) = \mathrm{poly}(kd/\epsilon)$. Given $M \in \mathbb{R}^{k_1 \cdots k_q \times d}$, the top $k$ singular vectors $U$ can be computed by computing the SVD of $M$, which is also done in time $\mathrm{poly}(kd/\epsilon)$. Once $U$ is obtained, the algorithm can terminate, which yields the desired runtime. $\qquad\square$

To complete the proof of the main theorem, we will need to prove Lemma D.4. To do this, we begin by introducing two definitions.

**Definition D.1.** *A random matrix $S$ is called a $\epsilon$-subspace embedding for a rank $k$ subspace $\mathcal{V}$ we have simultaneously for all $x \in \mathcal{V}$ that $\|Sx\|_2 = (1 \pm \epsilon)\|x\|_2$.*

**Definition D.2.** *A random matrix $S$ satisfies the $\epsilon$-approximate matrix product property if, for any fixed matrices $A, B$, of the appropriate dimensions, we have $\Pr[\|A^\top S^\top SB - A^\top B\|_F \leq \epsilon\|A\|_F\|B\|_F] \geq 9/10$.*

We now show that $S$ is both a subspace embedding and satisfies approximate matrix product, where $S = \otimes_{i=1}^{q} S_i$ and $S_i \in \mathbb{R}^{k_i \times n_i}$ are count-sketch matrices.

**Lemma D.3.** *If $S = (\otimes_{i=1}^{q} S_i)$ with $S_i \in \mathbb{R}^{k_i \times n_i}$, $k_1 = k_2 = \cdots = k_q = \Theta(q k^2/\epsilon^2)$, then $S$ is an $\epsilon$-subspace embedding for any fixed $k$ dimensional subspace $\mathcal{V} \subset \mathbb{R}^n$ with probability $9/10$, and also satisfies the $(\epsilon/k)$-approximate matrix product property.*

*Proof.* We first show that $S$ satisfies the $O(\epsilon/k, 1/10, 2)$-JL moment property. Here, the $(\epsilon, \delta, \ell)$-JL moment property means that for any fixed $x \in \mathbb{R}^n$ with $\|x\|_2 = 1$, we have $\mathbb{E}[(\|Sx\|_2^2 - 1)^2] \leq \epsilon^\ell \delta$, which will imply approximate matrix product by the results of [KN14].

We prove this by induction on $q$. Let $\bar{k} = k_1$. First suppose $S = (Q \otimes T)$, where $Q \in \mathbb{R}^{k_1 \times n_1}$ is a count-sketch, and $T \in \mathbb{R}^{k' \times n'}$ is any random matrix which satisfies $\mathbb{E}[\|Tx\|_2^2] = \|x\|_2^2$ ($T \in \mathbb{R}^{k' \times n'}$ is unbiased), and $\mathbb{E}[(\|Tx\|_2 - 1)^2] \leq 1 + c/\bar{k}$ for some value $c < \bar{k}$. Note that both of these properties are satisfied with $c = 4$ if $T \in \mathbb{R}^{k_2 \times n_2}$ is itself a count-sketch matrix [CW13]. Moreover, these are the only properties we will need about $T$, so we will. We now prove that $\mathbb{E}[\|(S \otimes T)x\|_2^2] = 1$ and $\mathbb{E}[\|(S \otimes T)x\|_2^4] \leq 1 + (c + 4)/\bar{k}$ for any unit vector $x$.

Fix any unit $x \in \mathbb{R}^n$ now (here $n = n_1 n'$), and let $x^j \in \mathbb{R}^{n'}$ be the vector obtained by restricted $x$ to the coordinates $j n_1 + 1$ to $(j+1)n_1$. For any $i \in [k_1], j \in [k']$, let $i_j = (i-1)k' + j$. Let $h_Q(i) \in [k_1]$ denote the row where the non-zero entry in the $i$-th column is placed in $Q$. Let $\sigma_Q(i) \in \{1, -1\}$

1100   denote the sign of the entry $Q_{h_Q(i),i}$. Let $\delta_Q(i,j)$ indicate the event that $h_Q(i) = j$. First note that

$$
\mathbb{E}\left[\sum_{i,j}((Q \otimes T)x)_{ij}^2\right] = \mathbb{E}\left[\sum_{i=1}^{k_1}\sum_{j=1}^{k'}\left(\sum_{\tau=1}^{n_1}\delta_Q(\tau,i)\sigma_Q(\tau)(Tx^\tau)_j\right)^2\right]
$$

$$
= \mathbb{E}\left[\sum_{i=1}^{k_1}\sum_{j=1}^{k'}\sum_{\tau=1}^{n_1}\delta_Q(\tau,i)(Tx^\tau)_j^2\right]
$$

$$
= \mathbb{E}\left[\sum_{\tau=1}^{n_1}\sum_{i=1}^{k_1}\sum_{j=1}^{k'}\delta_Q(\tau,i)(Tx^\tau)_j^2\right]
$$

$$
= \mathbb{E}\left[\sum_{\tau=1}^{n_1}\|Tx^\tau\|_2^2\right]
$$

$$
= \|x\|_2^2
$$

1101   Where the last equality follows because count-sketch $T$ is unbiased for the base case, namely that
1102   $\mathbb{E}[\|Tx\|_2^2] = \|x\|_2^2$ for any $x$ [Woo14], or by induction. We now compute the second moment,

$$
\mathbb{E}\left[\left(\sum_{i,j}((Q \otimes T)x)_{ij}^2\right)^2\right] = \mathbb{E}\left[\left(\sum_{i,j}\left(\sum_{\tau=1}^{n_1}\delta_Q(\tau,i)\sigma_Q(\tau)(Tx^\tau)_j\right)^2\right)^2\right]
$$

$$
= \mathbb{E}\left[\left(\sum_{i,j}\sum_{\tau_1,\tau_2}\delta_Q(\tau_1,i)\sigma_Q(\tau_1)(Tx^{\tau_1})_j\delta_Q(\tau_2,i)\sigma_Q(\tau_2)(Tx^{\tau_2})_j\right)^2\right]
$$

$$
= \sum_{\tau_1,\tau_2,\tau_3,\tau_4}\mathbb{E}\left[\left(\sum_{i,j}\delta_Q(\tau_1,i)\sigma_Q(\tau_1)(Tx^{\tau_1})_j\delta_Q(\tau_2,i)\sigma_Q(\tau_2)(Tx^{\tau_2})_j\right)\right.
$$

$$
\left. \cdot\left(\sum_{i,j}\delta_Q(\tau_3,i)\sigma_Q(\tau_3)(Tx^{\tau_3})_j\delta_Q(\tau_4,i)\sigma_Q(\tau_4)(Tx^{\tau_4})_j\right)\right] .
$$

1103   We now analyze the above expectation. There are several cases for the expectation of each term.
1104   First, we bound the sum of the expectations when $t_1 = t_2 = t_3 = t_4$ by

$$
\sum_{\tau=1}^{n_1}\mathbb{E}\left[\left(\sum_{i,j}\delta_Q(\tau,i)\sigma_Q(\tau)(Tx^\tau)_j\delta_Q(\tau,i)\sigma_Q(\tau)(Tx^\tau)_j\right)\right.
$$

$$
\left. \cdot\left(\sum_{i,j}\delta_Q(\tau,i)\sigma_Q(\tau)(Tx^\tau)_j\delta_Q(\tau,i)\sigma_Q(\tau)(Tx^\tau)_j\right)\right]
$$

$$
\leq \sum_{\tau=1}^{n_1}\mathbb{E}\left[\|Tx^\tau\|_2^4\right] = 1 + c/\bar{k}
$$

1105 Where the last equation follows from the variance of count-sketch [CW13] for the base case, or by
1106 induction for $q \geq 3$. We now bound the sum of the expectations when $t_1 = t_2 \neq t_3 = t_4$ by

$$\sum_{\tau_1 \neq \tau_2} \mathbb{E}\left[\left(\sum_{i,j} \delta_Q(\tau_1,i)\sigma_Q(\tau_1)(Tx^{\tau_1})_j \delta_Q(\tau_1,i)\sigma_Q(\tau_1)(Tx^{\tau_1})_j\right) \right.$$

$$\left. \cdot \left(\sum_{i,j} \delta_Q(\tau_2,i)\sigma_Q(\tau_2)(Tx^{\tau_2})_j \delta_Q(\tau_2,i)\sigma_Q(\tau_2)(Tx^{\tau_2})_j\right)\right]$$

$$\leq \sum_{\tau_1 \neq \tau_2} \mathbb{E}[\|Tx^{\tau_1}\|_2^2 \|Tx^{\tau_2}\|_2^2 / k_1]$$

$$\leq \mathbb{E}[\|Tx\|_2^4 / k_1] \leq (1 + c/\bar{k})/k_1.$$

We can similarly bound the sum of the terms with $t_1 = t_3 \neq t_2 = t_4$ and $t_1 = t_4 \neq t_3 = t_2$ by
$(1 + c/\bar{k})/k_1$, giving a total bound on the second moment of

$$\mathbb{E}[\|(Q \otimes T)x\|_2^4] \leq 1 + c/\bar{k} + 3(1 + c/\bar{k})/k_1) \leq 1 + (4+c)/\bar{k}$$

since any term with a $t_i \notin \{t_1, t_2, t_3, t_4\} \setminus \{t_i\}$ immediately has expectation 0. By induction, it
follows that $\mathbb{E}[(\otimes_{i=1}^q S_i)x\|_2^2] = 1$ for any unit $x$, and $\mathbb{E}[(\otimes_{i=1}^q S_i)x\|_2^4] \leq 1 + (4q+c)/\bar{k}$, where $c$ is
the constant from the original variance of count-sketch. Setting $\bar{k} = k_1 = \cdots = k_q = \Theta(qk^2/\epsilon^2)$
with a large enough constant, this completes the proof that $S = (\otimes_{i=1}^q S_i)$ has the $O(\epsilon/k, 1/10, 2)$-
JL moment property. Then by Theorem 21 of [KN14], we obtain the approximate matrix product
property:

$$\Pr[\|A^\top S^\top S B - A^\top B\|_F \leq O(\epsilon/k)\|A\|_F\|B\|_F] \geq 9/10$$

1107 for any two matrices $A, B$. Letting $A = B^\top = U$ where $U \in \mathbb{R}^{n \times k}$ is a orthogonal basis for any
1108 $k$-dimensional subspace $\mathcal{V} \subset \mathbb{R}^n$, it follows that

$$\|U^\top S^\top S U - I_k\|_F \leq O(\epsilon/k)\|U\|_F^2 \leq O(\epsilon),$$

1109 where the last step follows because $U$ is orthonormal, so $\|U\|_F^2 = k$. Since the Frobenius norm upper
1110 bounds the spectral norm $\|\cdot\|_2$, we have $\|U^\top S^\top S U - I_k\|_2 \leq O(\epsilon)$, from which it follows that all
1111 the eigenvalues of $U^\top S^\top S U$ are in $(1 - O(\epsilon), 1 + O(\epsilon))$, which implies $\|SUx\|_2 = (1 \pm O(\epsilon))\|x\|_2$
1112 for all $x \in \mathbb{R}^n$, so for any $y \in \mathcal{V}$, let $x_y$ be such that $y = Ux_y$, and then $\|Sy\|_2 = \|SUx_y\|_2 =$
1113 $(1 \pm O(\epsilon))\|x_y\|_2 = (1 \pm O(\epsilon))\|Ux_y\|_2 = (1 \pm O(\epsilon))\|y\|_2$, which proves that $S$ is a subspace
1114 embedding for $\mathcal{V}$ (not the second to last inequality holds because $U$ is orthonormal). □

1115 Finally, we are ready to prove Lemma D.4.

1116 **Lemma D.4.** *Let $S = (\otimes_{i=1}^q S_i)$ with $S_i \in \mathbb{R}^{k_i \times n_i}$, $k_1 = k_2 = \cdots = k_q = \Theta(qk^2/\epsilon^2)$. Then*
1117 *with probability $9/10$ $SA$ is a Projection Cost Preserving Sketch (PCP) for $A$, namely for all rank*
1118 *$k$ orthogonal projection matrix $P \in \mathbb{R}^{d \times d}$,*

$$(1 - \epsilon)\|A - AP\|_F^2 \leq \|SA - SAP\|_F^2 + c \leq (1 + \epsilon)\|A - AP\|_F^2$$

1119 *where $c \geq 0$ is some fixed constant independent of $P$ (but may depend on $A$ and $SA$).*

1120 *Proof.* To demonstrate that $SA$ is a PCP, we show that the conditions of Lemma 10 of [CEM$^+$15]
1121 hold, which imply this result. Our result follows directly from Theorem 12 of [CEM$^+$15]. Note
1122 that all that is needed (as discussed below the theorem) for the proof is that $S$ is an $\epsilon$-subspace
1123 embedding for a fixed $k$-dimensional subspaces, and that $S$ satisfies the $(\epsilon/\sqrt{k})$ approximate matrix
1124 product property. By Lemma D.3, we have both $\epsilon$-subspace embedding for $S$ as well as a stronger
1125 $(\epsilon/k)$ approximate matrix product property. Thus Theorem 12 holds for the random matrix $S$ when
1126 $k_1 = k_2 = \cdots = k_q = \Theta(qk^2/\epsilon^2)$, which completes the proof.

1127 □

 **E   Entry-wise Norm Low T-rank Approximation**

We now demonstrate our results for low $\operatorname{trank}$ approximation of arbitrary input matrices. Specifically, we study the following problem, defined in [VL00]: given $A \in \mathbb{R}^{n^2 \times n^2}$, the goal is to output a $\operatorname{trank}$-$k$ matrix $B \in \mathbb{R}^{n^2 \times n^2}$ such that

$$\|B - A\|_\xi \leq \alpha \cdot \operatorname{OPT}. \tag{1}$$

for some $\alpha \geq 1$, where $\operatorname{OPT} = \min_{\operatorname{trank} - k \ A'} \|A' - A\|_\xi$,, where the $\operatorname{trank}$ of a matrix $B$ is defined as the smallest integer $k$ such that $B$ can be written as a summation of $k$ matrices, where each matrix is the Kronecker product of $q$ matrices with dimensions $n \times n$: $B = \sum_{i=1}^k U_i \otimes V_i$, where $U_i, V_i \in \mathbb{R}^{n \times n}$.

Using Lemma B.2, we can rearrange the entries in $A \in \mathbb{R}^{n^2 \times n^2}$ to obtain $\overline{A} \in \mathbb{R}^{n^2 \times n^2}$, where the $(i + n(j-1))$'th row of $\bar{A}$ is equal to $vec((A_1)_{i,j} A_2)$, and also vectorize the matrix $U_i \in \mathbb{R}^{n \times n}$ and $V_i \in \mathbb{R}^{n \times n}$ to obtain vectors $u_i \in \mathbb{R}^{n^2}, v_i \in \mathbb{R}^{n^2}$. Therefore, for any entry-wise norm $\xi$ we have

$$\left\| \sum_{i=1}^k U_i \otimes V_i - A \right\|_\xi = \left\| \sum_{i=1}^k u_i v_i^\top - \overline{A} \right\|_\xi$$

**Lemma E.1** (Reshaping for Low Rank Approximation). *There is a one-to-one mapping $\pi : [n] \times [n] \times [n] \times [n] \to [n^2] \times [n^2]$ such that for any pairs $(U, u) \in \mathbb{R}^{n \times n} \times \mathbb{R}^{n^2}$ and $(V, v) \in \mathbb{R}^{n \times n} \times \mathbb{R}^{n^2}$, if $U_{i_1, j_1} = u_{i_1 + n(j_1 - 1)}$ and $V_{i_1, j_1} = v_{i_1 + n(j_1 - 1)}$, then we have for $i_1, i_2, j_1, j_2$*

$$(U \otimes V)_{i_1 + n(i_2 - 1), j_1 + n(j_2 - 1)} = (u \cdot v^\top)_{\pi(i_1, i_2, j_1, j_2)}$$

*where $U \otimes V \in \mathbb{R}^{n^2 \times n^2}$ and $uv^\top \in \mathbb{R}^{n^2 \times n^2}$.*

*Proof.* We have

$$\begin{aligned}
(U \otimes V)_{i_1 + n(i_2 - 1), j_1 + n(j_2 - 1)} &= U_{i_1, j_1} V_{i_2, j_2} \\
&= u_{i_1 + n(j_1 - 1)} \cdot v_{i_2 + n(j_2 - 1)} \\
&= (uv^\top)_{i_1 + n(j_1 - 1), i_2 + n(j_2 - 1)}
\end{aligned}$$

where the first step follows from the definition of $\otimes$ product, the second step follows from the connection between $U, V$ and $u, v$, the last step follows from the outer product. $\square$

Therefore, instead of using $\operatorname{trank}$ to define low-rank approximation of the $\otimes$ product of two matrices, we can just use the standard notion of rank to define it since both $B$ and $A'$ can be rearranged to have rank $k$.

**Definition E.2** (Based on Standard Notion of Rank). *Given two matrices $A_1, A_2 \times \mathbb{R}^{n \times n}$, let $\overline{A} \in \mathbb{R}^{n^2 \times n^2}$ denote the re-shaping of $A_1 \otimes A_2$. The goal is to output a rank-k matrix $\overline{B}$ such that*

$$\|\overline{B} - \overline{A}\|_\xi \leq \alpha \operatorname{OPT}_{\xi, k}$$

*where $\operatorname{OPT}_{\xi, k} = \min_{\operatorname{rank} - k \ \overline{A}'} \|\overline{A}' - \overline{A}\|_\xi$.*

In other words, $\overline{B}$ can be written as $\overline{B} = \sum_{i=1}^k u_i v_i^\top$ where $u_i, v_i$ are length $n^2$ vectors.

Combining the low-rank reshaping Lemma E.1 with the main input-sparsity low-rank approximation of [CW13], we obtain our Frobenius norm low rank approximation result.

**Theorem E.3** (Frobenius norm low rank approximation, $p = 2$). *For any $\epsilon \in (0, 1/2)$, there is an algorithm that runs in $n^2 \operatorname{poly}(k/\epsilon)$ and outputs a rank-k matrix $\overline{B}$ such that $\|\overline{B} - \overline{A}\|_F \leq (1 + \epsilon) \operatorname{OPT}_{F, k}$ holds with probability at least $9/10$, where $\operatorname{OPT}_p$ is cost achieved by best rank-k solution under the $\ell_p$-norm.*

Similarly, using the main $\ell_p$ low rank approximation algorithm of [SWZ17], we have

**Theorem E.4** (Entry-wise $\ell_p$-norm low rank approximation, $1 \leq p \leq 2$). *There is an algorithm that runs in $n^2 \operatorname{poly}(k)$ and outputs a rank-k matrix $\overline{B}$ such that $\|\overline{B} - \overline{A}\|_p \leq \operatorname{poly}(k \log n) \operatorname{OPT}_{p, k}$ holds with probability at least $9/10$, where $\operatorname{OPT}_p$ is cost achieved by best rank-k solution under the $\ell_p$-norm.*

1164    Applying the bi-criteria algorithm of [CGK$^+$17] gives us:

1165    **Theorem E.5** (General $p > 1$, bicriteria algorithm). *There is an algorithm that runs in* $\mathrm{poly}(n, k)$
1166    *and outputs a rank-*$\mathrm{poly}(k \log n)$ *matrix* $\overline{B}$ *such that* $\|\overline{B} - \overline{A}\|_p \leq \mathrm{poly}(k \log n) \, \mathrm{OPT}_p$ *holds with*
1167    *probability at least* $9/10$, *where* $\mathrm{OPT}_{p,k}$ *is cost achieved by best rank-$k$ solution under the $\ell_p$-norm.*

1168    Finally using the low-rank approximation algorithm for general loss functions given in [SWZ18],
1169    we obtain a very general result. The parameters for the loss function described in the following
1170    theorem are discussed in Section F.

1171    **Theorem E.6** (General loss function $g$). *For any function $g$ that satisfies Definition F.1, F.2, F.3,*
1172    *there is an algorithm that runs in* $O(n^2 \cdot T_{\mathrm{reg},g,n^2,k,n^2})$ *time and outputs a* $\mathrm{rank}$-$O(k \log n)$ *matrix*
1173    $\overline{B} \in \mathbb{R}^{n^2 \times n^2}$ *such that*

$$\|\overline{B} - \overline{A}\|_g \leq \mathrm{ati}_{g,k} \cdot \mathrm{mon}_g \cdot \mathrm{reg}_{g,k} \cdot O(k \log k) \cdot \mathrm{OPT}_{g,k},$$

1174    *holds with probability* $1 - 1/\mathrm{poly}(n)$.

1175    Hence, overall, the strategy is to first reshape $A = A_1 \otimes A_2$ into $\bar{A}$, then compute $\bar{B} = \sum_{i=1}^{k} u_i v_i^\top$
1176    using any of the above three theorems depending on the desired norm, and finally reshape $u_i$ and
1177    $v_i$ back to $U_i \in \mathbb{R}^{n \times n}$ and $V_i \in \mathbb{R}^{n \times n}$. It is easy to verify that the guarantees from Theo-
1178    rems E.6, E.4, E.3 are directly transferable to the guarantee of the $\mathrm{trank} - k$ approximation shown
1179    in Eq 1.

## F   Properties for General Loss Functions for Low trank Approximation

1181    We re-state three general properties (defined in [SWZ18]), the first two of which are structural
1182    properties and are necessary and sufficient for obtaining a good approximation from a small subset
1183    of columns. The third property is needed for efficient running time.

1184    **Definition F.1** (Approximate triangle inequality). *For any positive integer $n$, we say a function*
1185    $g(x) : \mathbb{R} \to \mathbb{R}_{\geq 0}$ *satisfies the* $\mathrm{ati}_{g,n}$-*approximate triangle inequality if for any* $x_1, x_2, \cdots, x_n \in \mathbb{R}$
1186    *we have*

$$g\left(\sum_{i=1}^{n} x_i\right) \leq \mathrm{ati}_{g,n} \cdot \sum_{i=1}^{n} g(x_i).$$

1187    **Definition F.2** (Monotone property). *For any parameter* $\mathrm{mon}_g \geq 1$, *we say function* $g(x) : \mathbb{R} \to$
1188    $\mathbb{R}_{\geq 0}$ *is* $\mathrm{mon}_g$-*monotone if for any* $x, y \in \mathbb{R}$ *with* $0 \leq |x| \leq |y|$, *we have* $g(x) \leq \mathrm{mon}_g \cdot g(y)$.

1189    **Definition F.3** (Regression property). *We say function* $g(x)$ : $\mathbb{R}$ $\to$ $\mathbb{R}_{\geq 0}$ *has the*
1190    $(\mathrm{reg}_{g,d}, T_{\mathrm{reg},g,n,d,m})$-*regression property if the following holds: given two matrices* $A \in \mathbb{R}^{n \times d}$
1191    *and* $B \in \mathbb{R}^{n \times m}$, *for each* $i \in [m]$, *let* $\mathrm{OPT}_i$ *denote* $\min_{x \in \mathbb{R}^d} \|Ax - B_i\|_g$. *There is an algorithm*
1192    *that runs in* $T_{\mathrm{reg},g,n,d,m}$ *time and outputs a matrix* $X' \in \mathbb{R}^{d \times m}$ *such that*

$$\|AX'_i - B\|_g \leq \mathrm{reg}_{g,d} \cdot \mathrm{OPT}_i, \forall i \in [m]$$

1193    *and outputs a vector* $v \in \mathbb{R}^d$ *such that*

$$\mathrm{OPT}_i \leq v_i \leq \mathrm{reg}_{g,d} \cdot \mathrm{OPT}_i, \forall i \in [m].$$

1194    *The success probability is at least* $1 - 1/\mathrm{poly}(nm)$.