[Reviews · NeurIPS 2019]

Reviewer 1



***** Post Rebuttal ****** The authors response did not change my evaluation. I strongly suggest to provide better motivation for the problem, and also discuss alternative approaches (e.g., first-order method)s. ***************************** The paper studies efficient algorithms for solving Kronecker Product Regression problems. These are interesting problems, since explicitly forming the regression problem and solving it requires typically huge amount of memory and runtime, but on the otherhand, the regression matrix is highly-structured and hence there is hope of having efficient solvers which do not require to explicitly "write down" the problem. Concretely, given matrices A_1,...A_q, each of dimension n_ixd_i, we want to solve L_p regression (1<=p<=2) with the matrix being the Kronecker product of A_1,...A_q (this is a matrix of dimension n_1n_2...n_qxd_1d_2...d_q) and the vector of outcomes b is of the corresponding dimension (n_1n_2...n_q). A recent result (DSSW18]) showed how to solve the most interesting (to my taste) case of $p=2$ with runtime that depends only on the sum of non-zero entries in the matrices A_1,...A_q (which is naturally unavoidable) and the number of non-zeros in the outcome vector b (which might be avoidable). Since b will typically be dense, this runtime is not ideal because of the huge dimension of b. Thus, the main question answered in the paper is to obtain an improved result of runtime that only depends on the overall number of non-zero entries in the input matrices A_1,...A_q. The authors provide improved results also for the case p<2 (although the improvement is less dramatic - still depends on nnz(b)) and for some related problems such as the-all-pairs regression problem. On the technical side, the authros claim to develop and use novel techniques for sampling from Kronecker products to construct a compact sketch and then solve it via standard methods (SVD for p=2). The authors also conduct empirical tests showing their method improves in runtime over the previous work. While I am certainly not an expert on subject (e.g., sketching techniques used), I think that from an algorithmic point of view, the problem is indeed interesting and challenging, and it seems the authors advance the state-of-the-art with a solid and clear contribution. Some issues: 1. from the references provided, it is not clear how much such regression problems are indeed interesting in terms of applications (or are they just theoretically interesting?) 2. In principle we can solve such problems with first-order methods for convex optimization. Is it clear that something like SGD are computing gradients in a "smart way" cannot give something better than explicitly forming the problem? I think such a discussion makes sense.

Reviewer 2



This paper gives randomized algorithms for Kronecker product regression problems. The claimed run-time of the method is a slight improvement over the state of the art in randomized algorithms. They consider L2 and Lp regression for p in [1,2]. The method is based on obtaining leverage scores of the Kronecker factors. The key observation is that approximate leverage score distribution of the factors yield an approximate leverage score distribution of the product. The algorithm subsamples the matrix proportional to the leverage scores and solves the smaller linear system via classical methods. The algorithm utilizes input sparsity time leverage score approximation methods based on the work by Clarkson and Woodruff, 2009. They also extend the result to Lp regression using p-stable random variables. For L2 regression, the run-time of the proposed method does not depend on the number of nonzeros of the right-hand-side vector, which is an improvement over previous run-time bounds when the vector is dense. Although the contributions appear to be sufficiently novel. I have some concerns on the hidden constants and the practicality of the method. Please see below for detailed comments. (1) In Theorem 3.1 and it's proof, is m=O(1 / (\delta \epsilon^2)) or m=O(d / (\delta \epsilon^2)) ? It looks like a factor of d is missing. (2) The run-time guarantee of Theorem 3.1 involves an unknown constant in the O(1) term, in O( \sum nnz(A_i) + (dq/\delta\epsilon))^(O(1)). Is there a bound on this term ? (3) The authors need to motivate problems involving extremely rectangular, i.e., n>> d, Kronecker product matrices. Otherwise, the run-time involving d^O(1) is not smaller compared to direct methods. (4) The authors claim that O(nnz(b)) is larger than O(\sum_i nnz(A_i)) to establish a faster run-time. Is this realistic ? This also needs to motivated. (5) Conjugate Gradient algorithm on well-conditioned linear systems Ax=b run in O(nnz(A)) time. Is there a similar guarantee for well-conditioned Kronecker product systems ?

Reviewer 3



The improvement of the complexity results stems from tensor reshaping described in Lemma B.2 so that the sketching for tensor can be reduced to that for matrices. It follows closely the work by Diao et al. AISTATS 2018. However, it seems non-trivial to piece together different components with details for achieving the results. Also, extensions with analysis to other related problems are considered. It would be good if experiments on real data can be given. Missing reference which might be related Li, "very sparse stable random projections", SIGKDD 2007

[Author Response · NeurIPS 2019]

We thank the reviewers for their detailed comments and suggestions. We will address the minor presentation comments in the paper, and will address the remaining comments below.

**Reviewer 1:** For the first reviewer, we would like to emphasize that while the case of $p = 2$ is central in practice, the case of $p = 1$ is also very important, and in fact is often more desirable than $p = 2$ for its robustness properties. For instance, in the rank regression estimator also considered in this paper, it is necessary that regression is carried out with respect to absolute deviation ($p = 1$), and not $p = 2$. For $p < 2$, our results demonstrate an even larger improvement over the prior work, which had runtime that grew super-linearly in $\sum_i \text{nnz}(A_i)$. Moreover, in several interesting cases where the vector $b$ itself is a Kronecker product (such as all-pairs regression), our $p < 2$ algorithms do not even require the $\text{nnz}(b)$ runtime, thus matching our results for $p = 2$. Additionally, regarding the remark made by reviewers 1 and 2 about a comparison of our work with SGD, we note that getting provable relative-error regression guarantees with first order methods such as SGD would indeed require a condition number assumption on the matrices $A_1, \ldots, A_q$ (note that the condition number of $\otimes_{i=1}^q A_i$ is the product of the condition numbers of $A_i$). Our algorithms, interestingly, do not depend at all on the condition number of the $A_i$'s, and make no assumptions on these matrices.

**Reviewer 2** In response to the reviewer's worries about practicality and applicability of our results, we would like to reiterate that Kronecker products do arise naturally in many applications. Significantly, Kronecker product regression comes up very frequently in tensor related-problems, which are relevant whenever the data has more than two associated dimensions. For instance, tensor regression is used centrally in [1] as well as in [2]. Kronecker products naturally arise when solving matrix equations, which show up in many different settings (see e.g. [4]). For instance, solving $AX - XB = C$, which is the Sylvester equation, can be written as $(A \otimes B)\text{vec}(x) = \text{vec}(c)$. Additionally, Kronecker products of more than two matrices arise when looking at partial differential equations such as a Poisson equation. Another important motivation to study Kronecker product regression is that a common way of solving low rank approximation (LRA) is via alternating minimization. Namely, if you want a LRA to a matrix $A \in \mathbb{R}^{n \times n}$, you can first fix some $U \in \mathbb{R}^{n \times k}$ and solve for the $V \in \mathbb{R}^{k \times n}$ that minimizes $\|A - UV\|$ via regression. Once you have a $V$, then you solve for the best $U$ given the $V$, and so on. The same procedure is used for third-order tensor low rank approximation, where you have $A \approx U \otimes V \otimes W$ and use regression to solve for one of the factors at a time with the other two fixed, which is now a Kronecker product regression problem.

With regards to dependence on $d$ in Theorem 3.1, we are indeed missing a factor of $d$ in the definition of $m$ – we will fix this typo. With respect to the reviewer's comment on the run-time of this theorem, we first note that the main computation taking place is the leverage score computation from Proposition A.3. For a constant number of input matrices $q$ (as is generally the case in applications), the term $d^{O(1)}$ in Proposition A.3 to approximate leverage scores is $O(d^3)$. The remaining computation is to compute the pseudo-inverse of a $d/\epsilon^2 \times d$ matrix, which requires $O(d^3/\epsilon^2)$ time, so the additive term in Theorem 3.1 can be replaced with $O(d^3/\epsilon^2)$. This implies that even for $q = 2$ matrices, if $n > d^2$, the size of the Kronecker product will be $\Omega(d^4) \times d$, which is larger than the runtime of our algorithm.

Additionally, the regime where $n \gg d$, known as the over-constrained regime, is motivated by many common situations in practice, and as a result has been the focus of a large amount of algorithmic research over the past decade (as cited in our submission). The special case of Kronecker products is no different, and also frequently arises in the $n \gg d$ setting, such as [3] which considers very rectangular matrices (see their experiments in Table 5.1). This setting addresses another comment of the second reviewer regarding the relative sizes of $\text{nnz}(b)$, and $\sum_i \text{nnz}(A_i)$. Even in the above example with $n$ only slightly larger than $d$, we would have $\text{nnz}(b) = \Omega(d^4)$ but our runtime would be $O(d^3)$. The above shows that our algorithms become much more practical than both traditional linear algebraic algorithms and SGD (which can have a bad dependence on the condition number) even in a mildly over-constrained setting.

**Reviewer 3** With regards to the comment on the impact of our algorithms in applications, we first point to the above paragraphs which discuss settings in practice where our algorithms have been proven to perform substantially better than prior techniques. In addition, we remark that our experiments section demonstrates strong improvements on the run-time of our algorithms when compared to the prior work of Diao et al. This suggests that our algorithms are even faster than the provable guarantee of $O(d^3)$, which may be pessimistic because it is a worst-case theoretical bound.

[1] O. A. Malik and S. Becker. Low-rank tucker decomposition of large tensors using tensorsketch. In *Advances in Neural Information Processing Systems*, pages 10096–10106, 2018.

[2] O. A. Malik and S. Becker. Fast randomized matrix and tensor interpolative decomposition using countsketch. *arXiv preprint arXiv:1901.10559*, 2019.

[3] G. Pisinger and A. Zimmermann. Linear least squares problems with data over incomplete grids. *BIT Numerical Mathematics*, 47(4):809–824, 2007.

[4] V. Simoncini. Computational methods for linear matrix equations. *SIAM Review*, 58(3):377–441, 2016.


[Meta-Review · NeurIPS 2019]

This is a borderline paper. The reviewers suggest, and I agree with them, that the paper can be improved by a convincing real world application, stronger motivation of the problem, discussion of alternative approaches, and a more thorough discussion of practical aspects of the algorithm.